# A novel ATP dependent dimethylsulfoniopropionate lyase in bacteria that releases dimethyl sulfide and acryloyl-CoA

**Chun-Yang Li[1,2†], Xiu-Juan Wang[1†], Xiu-Lan Chen[1,3], Qi Sheng[1], Shan Zhang[1], Peng Wang[2], Mussa Quareshy[4], Branko Rihtman[4], Xuan Shao[1], Chao Gao[1], Fuchuan Li[5], Shengying Li[1], Weipeng Zhang[2], Xiao-Hua Zhang[2], Gui-Peng Yang[6], Jonathan D Todd[7], Yin Chen[2,4], Yu-Zhong Zhang[2,3,8]***

[1]State Key Lab of Microbial Technology, Marine Biotechnology Research Center, Shandong University, Qingdao, China; [2]College of Marine Life Sciences, Ocean University of China, Qingdao, China; [3]Laboratory for Marine Biology and Biotechnology, Pilot National Laboratory for Marine Science and Technology, Qingdao, China; [4]School of Life Sciences, University of Warwick, Coventry, United Kingdom; [5]National Glycoengineering Research Center and Shandong Key Laboratory of Carbohydrate Chemistry and Glycobiology, Shandong University, Qingdao, China; [6]Frontiers Science Center for Deep Ocean Multispheres and Earth System, Key Laboratory of Marine Chemistry Theory and Technology, Ministry of Education, Ocean University of China, Qingdao, China; [7]School of Biological Sciences, University of East Anglia, Norwich Research Park, Norwich, United Kingdom; [8]Marine Biotechnology Research Center, State Key Laboratory of Microbial Technology, Shandong University, Qingdao, China

**\*For correspondence:**
zhangyz@sdu.edu.cn

† Chun-Yang Li and Xiu-Juan Wang contributed equally to this work.

**Competing interest:** The authors declare that no competing interests exist.

**Abstract** Dimethylsulfoniopropionate (DMSP) is an abundant and ubiquitous organosulfur molecule in marine environments with important roles in global sulfur and nutrient cycling. Diverse DMSP lyases in some algae, bacteria, and fungi cleave DMSP to yield gaseous dimethyl sulfide (DMS), an infochemical with important roles in atmospheric chemistry. Here, we identified a novel ATP-dependent DMSP lyase, DddX. DddX belongs to the acyl-CoA synthetase superfamily and is distinct from the eight other known DMSP lyases. DddX catalyses the conversion of DMSP to DMS via a two-step reaction: the ligation of DMSP with CoA to form the intermediate DMSP-CoA, which is then cleaved to DMS and acryloyl-CoA. The novel catalytic mechanism was elucidated by structural and biochemical analyses. DddX is found in several Alphaproteobacteria, Gammaproteobacteria, and Firmicutes, suggesting that this new DMSP lyase may play an overlooked role in DMSP/DMS cycles.

## Introduction

The organosulfur molecule dimethylsulfoniopropionate (DMSP) is produced in massive amounts by many marine phytoplankton, macroalgae, angiosperms, bacteria, and animals (*Curson et al., 2018*; *Stefels, 2000*; *Otte et al., 2004*; *Curson et al., 2017*; *Raina et al., 2013*). DMSP can function as an antioxidant, osmoprotectant, predator deterrent, cryoprotectant, protectant against hydrostatic pressure, chemoattractant and may enhance the production of quorum-sensing molecules (*Sunda et al.,*

**eLife digest** The global sulfur cycle is a collection of geological and biological processes that circulate sulfur-containing compounds through the oceans, rocks and atmosphere. Sulfur itself is essential for life and important for plant growth, hence its widespread use in fertilizers.

Marine organisms such as bacteria, algae and phytoplankton produce one particular sulfur compound, called dimethylsulfoniopropionate, or DMSP, in massive amounts. DMSP made in the oceans gets readily converted into a gas called dimethyl sulfide (DMS), which is the largest natural source of sulfur entering the atmosphere. In the air, DMS is converted to sulfate and other by-products that can act as cloud condensation nuclei, which, as the name suggests, are involved in cloud formation. In this way, DMS can influence weather and climate, so it is often referred to as 'climate-active' gas.

At least eight enzymes are known to cleave DMSP into DMS gas with a few by-products. These enzymes are found in algae, bacteria and fungi, and are referred to as lyases, for the way they break-down their target compounds (DMSP, in this case). Recently, researchers have identified some bacteria that produce DMS from DMSP without using known DMSP lyases. This suggests there are other, unidentified enzymes that act on DMSP in nature, and likely contribute to global sulfur cycling.

Li, Wang et al. set out to uncover new enzymes responsible for converting the DMSP that marine bacteria produce into gaseous DMS. One new enzyme called DddX was identified and found to belong to a superfamily of enzymes quite separate to other known DMSP lyases. Li, Wang et al. also showed how DddX drives the conversion of DMSP to DMS in a two-step reaction, and that the enzyme is found across several classes of bacteria. Further experiments to characterise the protein structure of DddX also revealed the molecular mechanism for its catalytic action.

This study offers important insights into how marine bacteria generate the climatically important gas DMS from DMSP, leading to a better understanding of the global sulfur cycle. It gives microbial ecologists a more comprehensive perspective of these environmental processes, and provides biochemists with data on a family of enzymes not previously known to act on sulfur-containing compounds.

---

*2002*; *Cosquer et al., 1999*; *Wolfe et al., 1997*; *Karsten et al., 1996*; *Zheng et al., 2020*; *Seymour et al., 2010*; *Johnson et al., 2016*). DMSP also has important roles in global sulfur and nutrient cycling (*Kiene et al., 2000*; *Charlson et al., 1987*). Environmental DMSP can be taken up and catabolised as a carbon and/or sulfur source by diverse microbes, particularly bacteria (*Curson et al., 2011b*). DMSP catabolism can release volatile dimethyl sulfide (DMS) and/or methanethiol (MeSH) (*Reisch et al., 2011a*). DMS is a potent foraging cue for diverse organisms (*Nevitt, 2011*) and the primary biological source of sulfur transferred from oceans to the atmosphere (*Andreae, 1990*), which may participate in the formation of cloud condensation nuclei, and influence the global climate (*Vallina and Simó, 2007*).

Bacteria can metabolize DMSP via three known pathways, the demethylation pathway (*Howard et al., 2006*), the recently reported oxidation pathway (*Thume et al., 2018*), and the lysis pathway (*Curson et al., 2011b*; *Figure 1*). The nomenclature of these pathways is based on the reaction type of the enzyme catalyzing the first step of DMSP catabolism. In the demethylation pathway, DMSP demethylase DmdA first demethylates DMSP to produce methylmercaptopropionate (MMPA) (*Howard et al., 2006*; *Reisch et al., 2008*), which can be further catabolized to MeSH and acetaldehyde (*Figure 1*; *Reisch et al., 2011b*; *Bullock et al., 2017*; *Shao et al., 2019*). In the oxidation pathway, DMSP is oxidized to dimethylsulfoxonium propionate (DMSOP), which is further metabolized to dimethylsulfoxide (DMSO) and acrylate; however, enzymes involved in this pathway are unknown (*Thume et al., 2018*; *Figure 1*).

In the lysis pathway, diverse lyases cleave DMSP to produce DMS and acrylate or 3-hydroxypropionate-CoA (3-HP-CoA), which are further metabolized by ancillary enzymes (*Curson et al., 2011b*; *Johnston et al., 2016*; *Figure 1*). There is large biodiversity in DMSP lysis, with eight different known DMSP lyases that encompass four distinct protein families (DddD a CoA-transferase; DddP a metallopeptidase; cupin containing DddL, DddQ, DddW, DddK, and DddY; and Alma1 an aspartate racemase) functioning in diverse marine bacteria, algae, and fungi (*Figure 1*; *Curson et al., 2011b*; *Johnston et al., 2016*). With the exception of DddD, which catalyzes an acetyl-CoA-dependent

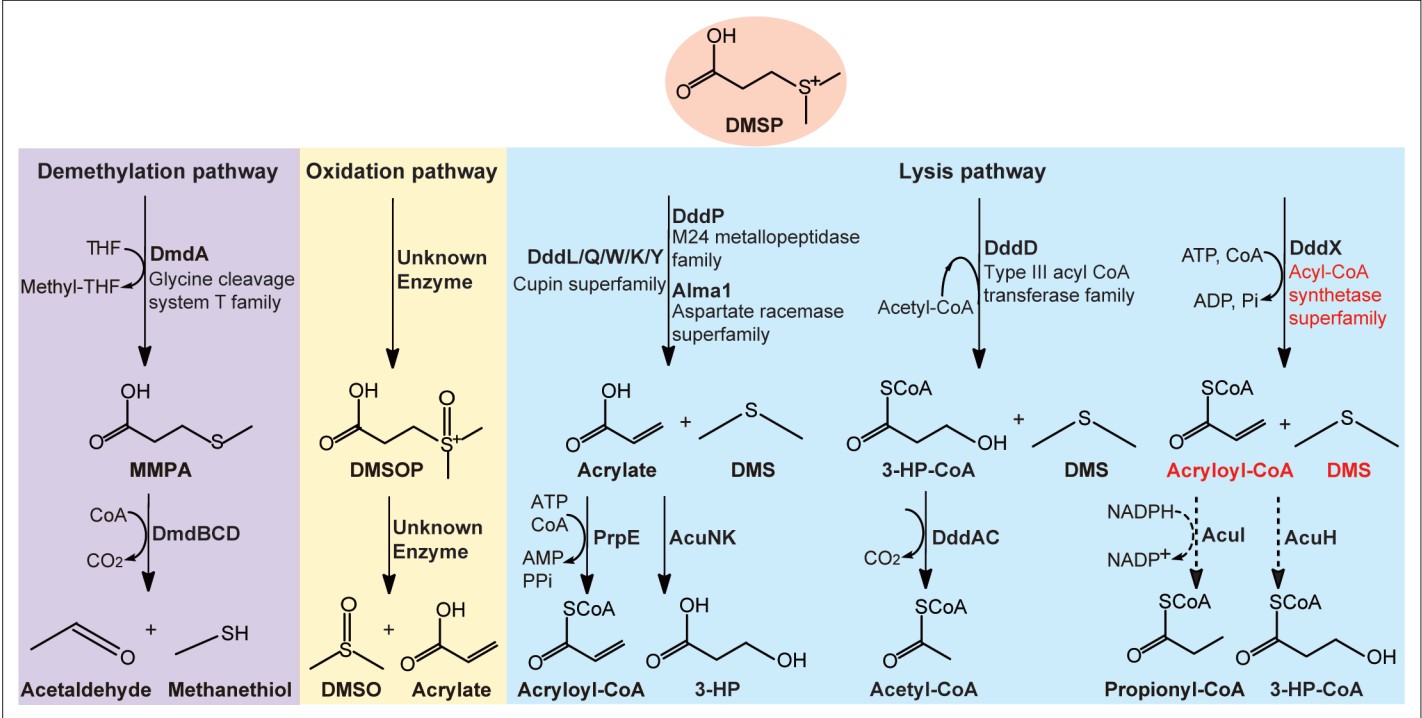

**Figure 1.** Metabolic pathways for DMSP degradation. Different pathways are shown in different colors. The demethylation of DMSP by DmdA produces MMPA (in purple). The oxidation of DMSP produces DMSOP (in yellow). In the lysis pathway (in blue), DMSP lyase DddP, DddL, DddQ, DddW, DddK, DddY, or Alma1 converts DMSP to acrylate and DMS, DddD converts DMSP to 3-HP-CoA and DMS, using acetyl-CoA as a CoA donor, and the newly identified DddX in this study converts DMSP to acryloyl-CoA and DMS, with ATP and CoA as co-substrates. Dotted lines represent unconfirmed steps of the DddX DMSP lysis pathway that we propose in this study. The protein families of enzymes involved in the first step of each pathway are indicated. The protein family of DddX and the products of its catalysis are highlighted in red color. THF, tetrahydrofolate; MMPA, methylmercaptopropionate; 3-HP, 3-hydroxypropionate; DMSOP, dimethylsulfoxonium propionate; DMSO, dimethylsulfoxide.

The online version of this article includes the following figure supplement(s) for figure 1:

**Figure supplement 1.** Enzymatic activity analysis of the recombinant DddX using acetyl-CoA as a CoA donor.

**Figure supplement 2.** HPLC assay of the enzymatic activity of 0105 protein on acryloyl-CoA at 260 nm.

CoA transfer reaction, all other DMSP lyases directly cleave DMSP (*Bullock et al., 2017*; *Todd et al., 2007*; *Alcolombri et al., 2014*; *Lei et al., 2018*; *Li et al., 2014*). Recently, several bacterial isolates were reported to produce DMS from DMSP but lack known DMSP lyases in their genomes (*Liu et al., 2018*; *Zhang et al., 2019*), suggesting the presence of novel enzyme(s) for DMSP degradation in nature.

A common feature of previously characterized DMSP metabolic pathways is that the metabolites (*i.e.* MMPA, acrylate) need to be ligated with CoA for further catabolism (*Figure 1*; *Curson et al., 2011b*; *Reisch et al., 2011b*). Currently, there is no known pathway whereby DMSP is ligated with free CoA, and it is tempting to speculate that there may be such a novel DMSP metabolic pathway. In this study, we screened DMSP-catabolizing bacteria from Antarctic samples, and obtained a strain *Psychrobacter* sp. D2 that grew on DMSP and produced DMS. Genetic and biochemical work showed that *Psychrobacter* sp. D2 possesses a novel DMSP lyase termed DddX for DMSP catabolism (*Figure 1*). DddX is an ATP-dependent DMSP lyase which catalyzes a two-step reaction: the ligation of DMSP and CoA, and the cleavage of DMSP-CoA to produce DMS and acryloyl-CoA. We further solved the crystal structure of DddX and elucidated the molecular mechanism for its catalysis based on structural and biochemical analyses. DddX is found in both Gram-negative and Gram-positive bacteria. Our results provide novel insights into the microbial metabolism of DMSP by this novel enzyme.

## Results

### A potentially novel DMSP lyase in a conventional DMSP catabolic gene cluster

Using DMSP (5 mM) as the sole carbon source, DMSP-catabolizing bacteria were isolated from five Antarctic samples including alga, sediments, and seawaters (*Figure 2—figure supplement 1*,

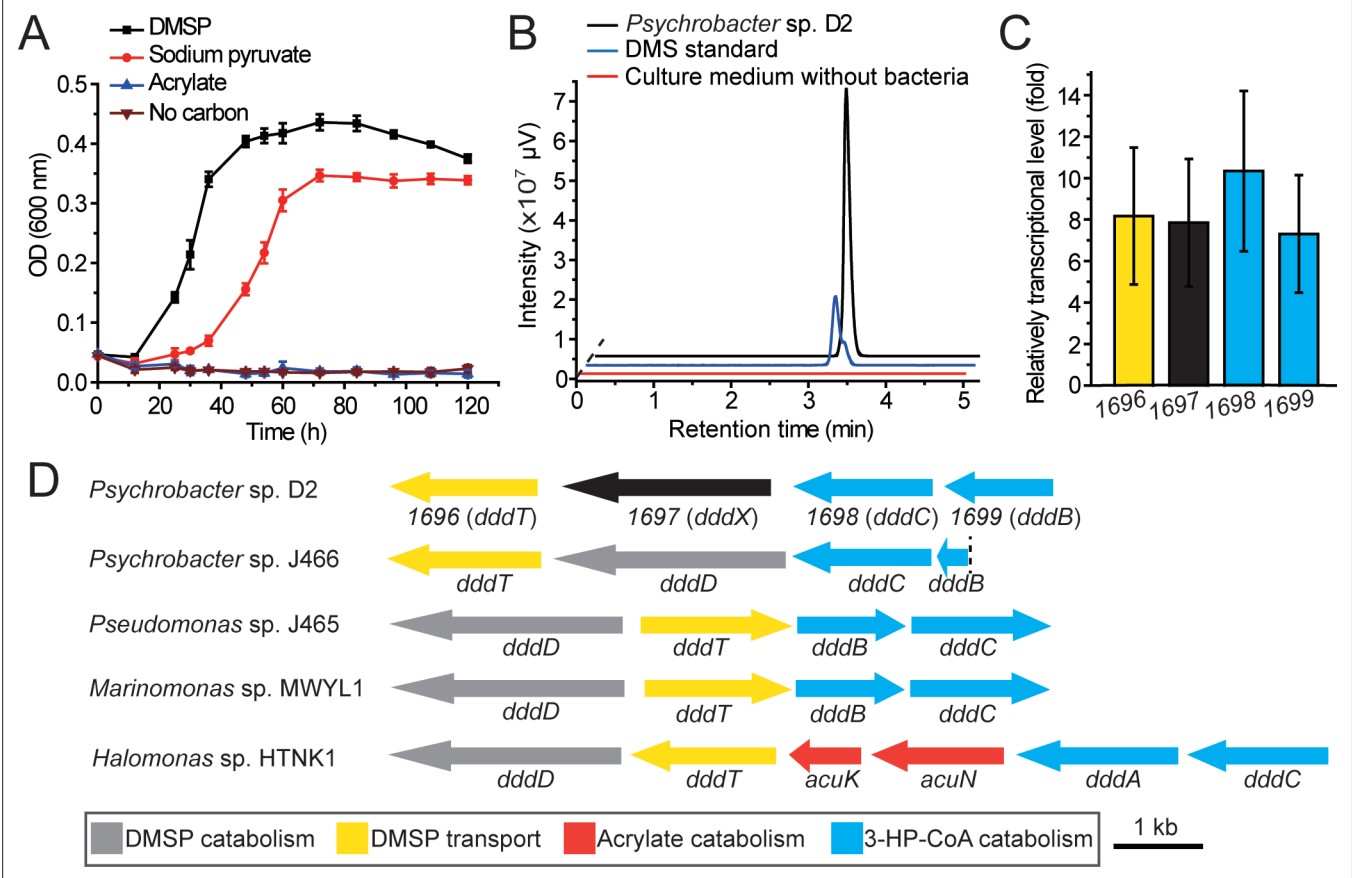

**Figure 2.** The utilization of DMSP by *Psychrobacter* sp. D2 and the putative DMSP-catabolizing. gene cluster in its genome. (**A**) The growth curve of *Psychrobacter* sp. D2 on DMSP, sodium pyruvate or acrylate as sole carbon source (5 mM) at 15°C. The error bar represents standard deviation of triplicate experiments. (**B**), GC detection of DMS production from DMSP by strain D2. The culture medium without bacteria was used as the control. The DMS standard was used as a positive control. *Psychrobacter* sp. D2 could catabolize DMSP and produce DMS (44.8 ± 1.8 nmol DMS min$^{-1}$ mg protein$^{-1}$). (**C**), RT-qPCR assay of the transcriptions of the genes *1696*, *1697*, 1,698, and 1,699 in *Psychrobacter* sp. D2 in response to DMSP in the marine broth 2,216 medium. The bacterium cultured without DMSP in the same medium was used as the control. The *recA* gene was used as an internal reference. The error bar represents standard deviation of triplicate experiments. The locus tags of *1696*, *1697*, 1,698, and 1,699 are H0262_08195, H0262_08200, H0262_08205, and H0262_08210, respectively. (**D**), Genetic organization of the putative DMSP-catabolizing gene cluster. Reported DMSP catabolic/ transport gene clusters from *Psychrobacter* sp. J466, *Pseudomonas* sp. J465, *Marinomonas* sp. MWYL1, and *Halomonas* sp. HTNK1 are shown (***Todd et al., 2007***; ***Todd et al., 2010***; ***Curson et al., 2010***; ***Curson et al., 2011b***). The dashed vertical line indicates a breakpoint in *dddB* in the cosmid library of *Pseudomonas* sp. J466 (***Curson et al., 2010***).

The online version of this article includes the following figure supplement(s) for figure 2:

**Source data 1.** The growth curve of *Psychrobacter* sp. D2 on DMSP, sodium pyruvate or acrylate as sole carbon source.

**Source data 2.** GC detection of DMS production from DMSP by strain D2.

**Source data 3.** RT-qPCR assay of the transcriptions of the genes *1696*, *1697*, 1,698, and 1,699 in *Psychrobacter* sp. D2.

**Figure supplement 1.** Locations of the sampling sites and the relative abundance of DMSP-catabolizing bacteria isolated from the samples.

**Figure supplement 1—source data 1.** The number of DMSP-catabolizing strains isolated from the Antarctic samples.

**Figure supplement 2.** Transcriptomic analysis of the putative genes involved in DMSP metabolism in strain D2.

**Figure supplement 2—source data 1.** Transcriptomic analysis of the putative genes involved in DMSP metabolism in strain D2.

**Figure supplement 3.** Confirmation of the deletion of the *dddX* gene from *Psychrobacter* sp.

*Supplementary file 1a*). In total, 175 bacterial strains were obtained (*Figure 2—figure supplement 1B*). Among these bacterial strains, *Psychrobacter* sp. D2, a marine gammaproteobacterium, grew well in the medium containing DMSP as the sole carbon source, but not acrylate (*Figure 2A*). Moreover, gas chromatography (GC) analysis showed that *Psychrobacter* sp. D2 could catabolize DMSP and produce DMS (44.8 ± 1.8 nmol DMS min$^{-1}$ mg protein$^{-1}$) (*Figure 2B*).

To identify the genes involved in DMSP degradation in *Psychrobacter* sp. D2, we sequenced its genome and searched homologs of known DMSP lyases. However, no homologs of known DMSP lyases with amino acid sequence identity higher than 30% were found in its genome (*Supplementary file 1b*), implying that this strain may possess a novel enzyme or a novel pathway for DMSP catabolism. We then sequenced the transcriptomes of this strain when grown with and without DMSP as the sole carbon source. Transcriptional data analyses showed that the transcripts of four genes (*1696*, *1697*, *1,698*, and *1699*) that compose a gene cluster were all highly upregulated (*Figure 2—figure supplement 2*) when DMSP was supplied as the sole carbon source, which was further confirmed by RT-qPCR analysis (*Figure 2C*). These results suggest that this gene cluster may participate in DMSP catabolism within *Psychrobacter* sp. D2.

In the gene cluster, 1696 is annotated as a betaine-carnitine-choline transporter (BCCT), sharing 32% amino acid identity with DddT, the predicted DMSP transporter in *Marinomonas* sp. MWYL1 (*Sun et al., 2012*; *Todd et al., 2007*); 1,697 is annotated as an acetate-CoA ligase, and shares 26% sequence identity with the acetyl-CoA synthetase (ACS) in *Giardia lamblia* (*Sánchez et al., 2000*); 1,698 is annotated as an aldehyde dehydrogenase, sharing 72% sequence identity with DddC in *Marinomonas* sp. MWYL1 (*Todd et al., 2007*); and 1,699 is annotated as an alcohol dehydrogenase, sharing 65% sequence identity with DddB in *Marinomonas* sp. MWYL1 (*Todd et al., 2007*). DddT, DddC, and DddB have been reported to be involved in DMSP import and catabolism (*Sun et al., 2012*; *Todd et al., 2007*; *Todd et al., 2010*). The pattern of the identified gene cluster *1696–1699* in *Psychrobacter* sp. D2 is similar to the patterns of those DMSP-catabolizing clusters reported in *Pseudomonas*, *Marinomonas,* and *Halomonas*, in which *dddT, dddB* and *dddC* are clustered with the DMSP lyase gene *dddD*, but which is missing in *1696–1699* and is replaced by *1697* (*Todd et al., 2007*; *Todd et al., 2010*; *Curson et al., 2010*; *Figure 2D*). These data further support that the *1696–1699* gene cluster is involved in *Psychrobacter* sp. D2 DMSP catabolism and *1697* encodes a DMSP lyase equivalent to DddD. However, the sequence identity between 1,697 and DddD is less than 15%, suggesting that 1,697 is unlikely a DddD homolog. With these data we predicted that 1,697 encodes a novel DMSP lyase in *Psychrobacter* sp. D2, which we term as DddX hereafter.

## The essential role of DddX in DMSP degradation in *Psychrobacter* Sp. D2

To identify the possible function of *dddX* in DMSP catabolism, we first deleted the majority of the *dddX* gene within the *Psychrobacter* sp. D2 genome to generate a Δ*dddX* mutant strain (*Figure 2—figure supplement 3*). The Δ*dddX* mutant was unable to grow on DMSP as the sole carbon source, but its ability to utilize DMSP was fully restored to wild type levels by cloned of *dddX* (in pBBR1MCS-*dddX*) (*Figure 3A*), indicating that *dddX* is essential for strain D2 to utilize DMSP. Furthermore, the Δ*dddX* mutant lost DMSP lyase activity, that is it no longer produced DMS when cultured in marine broth 2,216 medium with DMSP. DMSP lyase activity was fully restored to wild type levels in the complemented strain (Δ*dddX*/pBBR1MCS-*dddX*) (*Figure 3B*), indicating that *dddX* encodes a functional DMSP lyase enzyme degrading DMSP to DMS.

## DddX is an ATP-dependent DMSP lyase and its kinetic analysis

To verify the enzymatic activity of DddX on DMSP, we cloned the *dddX* gene, overexpressed it in *Escherichia coli* BL21 (DE3), and purified the recombinant DddX (*Figure 3—figure supplement 1*). Sequence analysis suggests that DddX is an acetate-CoA ligase, which belongs to the acyl-CoA synthetase (ACD) superfamily and requires CoA and ATP as co-substrates for catalysis (*Musfeldt and Schonheit, 2002*; *Mai and Adams, 1996*). Thus, we added CoA and ATP into the reaction system when measuring the enzymatic activity of the recombinant DddX on DMSP. GC analysis showed that the recombinant DddX directly acted on DMSP and produce DMS (*Figure 3C*). HPLC analysis uncovered ADP and an unknown product as DMS co-products (*Figure 3D*). The chromatographic retention time of the unknown product was consistent with it being acryloyl-CoA (*Wang et al., 2017*; *Cao*

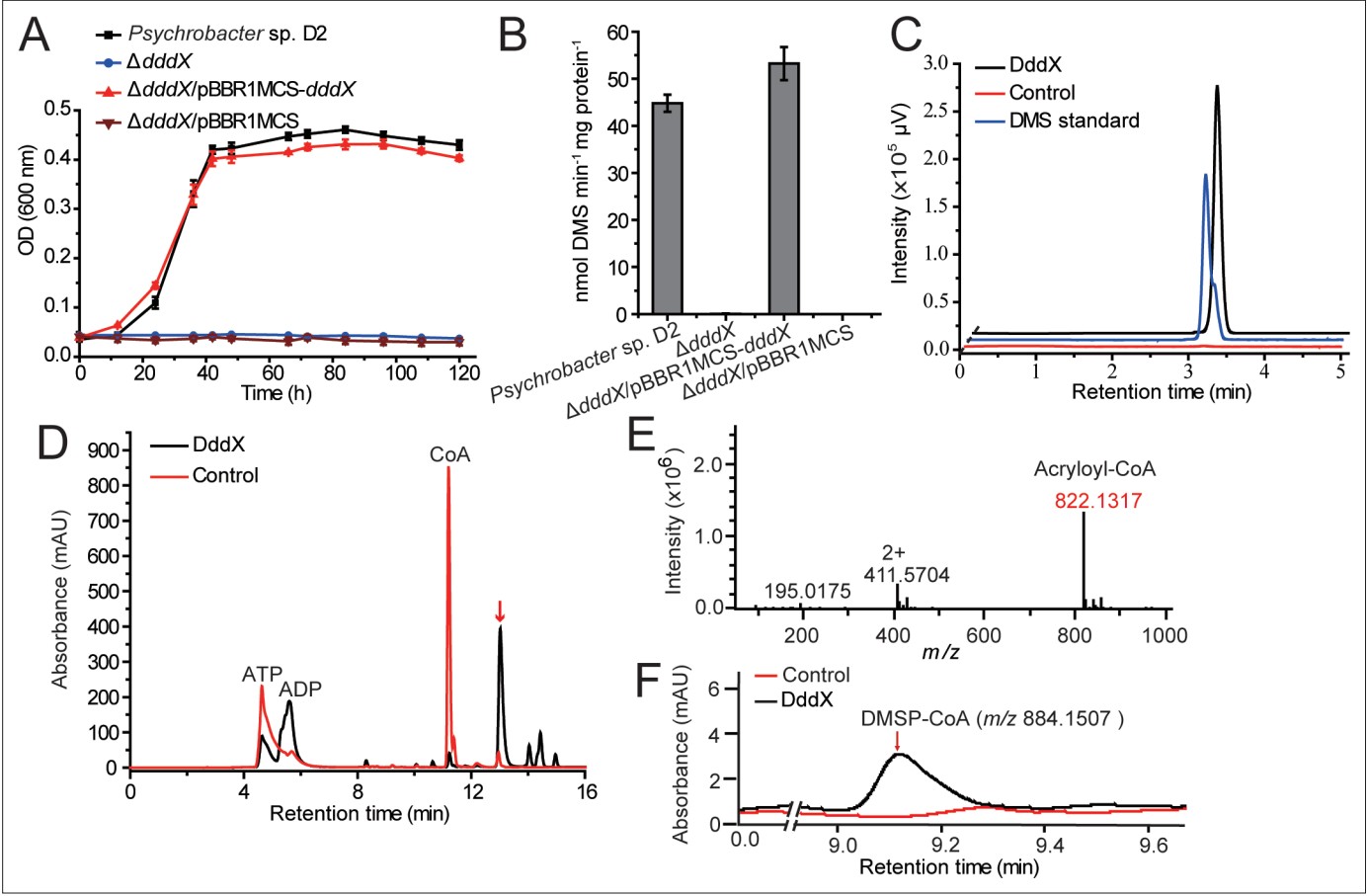

**Figure 3.** The function of *Psychrobacter* sp. D2 *dddX* in DMSP metabolism. (**A**) Growth curves of the wild-type strain D2, the Δ*dddX* mutant, the complemented mutant (Δ*dddX*/pBBR1MCS-*dddX*), and the Δ*dddX* mutant complemented with an empty vector (Δ*dddX*/pBBR1MCS). All strains were grown with DMSP (5 mM) as the sole carbon source. The error bar represents standard deviation of triplicate experiments. (**B**), Detection of DMS production from DMSP degradation by the wild-type strain D2, the Δ*dddX* mutant, the complemented mutant Δ*dddX*/pBBR1MCS-*dddX*, and the mutant complimented with an empty vector Δ*dddX*/pBBR1MCS. The error bar represents standard deviation of triplicate experiments. (**C**), GC detection of DMS production from DMSP lysis catalyzed by the recombinant DddX. The reaction system without DddX was used as the control. DddX maintained a specific activity of ~8.0 μmol min$^{-1}$ mg protein$^{-1}$ at 20°C, pH 8.0.( **D**), HPLC analysis of the enzymatic activity of the recombinant DddX on DMSP at 260 nm. The peak of the unknown product is indicated with a red arrow. The reaction system without DddX was used as the control. (**E**), LC-MS analysis of the unknown product. (**F**), HPLC analysis of the intermediate of DddX catalysis at 260 nm. The HPLC system was coupled to a mass spectrometer for *m/z* determination. The reaction system without DddX was used as the control.

The online version of this article includes the following figure supplement(s) for figure 3:

**Source data 1.** Growth curves of the wild-type strain D2, the Δ*dddX* mutant, the complemented mutant (Δ*dddX*/pBBR1MCS-*dddX*), and the Δ*dddX* mutant complemented with an empty vector (Δ*dddX*/pBBR1MCS).

**Source data 2.** Detection of DMS production from DMSP degradation by the wild-type strain D2, the Δ*dddX* mutant, the complemented mutant Δ*dddX*/pBBR1MCS-*dddX*, and the mutant complimented with an empty vector Δ*dddX*/pBBR1MCS.

**Source data 3.** GC detection of DMS production from DMSP lysis catalyzed by the recombinant DddX.

**Source data 4.** HPLC analysis of the enzymatic activity of the recombinant DddX on DMSP.

**Figure supplement 1.** SDS-PAGE analysis of the recombinant DddX.

**Figure supplement 2.** Two alternative mechanisms for DMSP degradation catalyzed by DddX.

**Figure supplement 3.** Characterization of recombinant DddX.

**Figure supplement 3—source data 1.** Characterization of recombinant DddX.

**Figure supplement 4.** HPLC assay of the enzymatic activity of DddX toward DMSP, sodium acetate, and sodium propionate at 260 nm.

**Figure supplement 5.** The effects of potential inhibitors on the enzymatic activity of DddX.

**Figure supplement 5—source data 1.** The effects of potential inhibitors on the enzymatic activity of DddX.

*Figure 3 continued on next page*

*Figure 3 continued*

**Figure supplement 6.** The growth curves of *Psychrobacter* sp.

**Figure supplement 6—source data 1.** The growth curves of *Psychrobacter* sp.

*et al., 2017*). Indeed, liquid chromatography-mass spectrometry (LC-MS) analysis found the molecular weight (MW) of the unknown product to be 822.1317, exactly matching acryloyl-CoA (*Figure 3E*). These data demonstrate that DddX is a functional ATP-dependent DMSP lyase that can catalyze DMSP degradation to DMS and acryloyl-CoA.

The biochemical results above suggest that DddX catalyzes a two-step degradation of DMSP, a CoA ligation reaction and a cleavage reaction. To perform this two-step reaction, there are two alternative pathways: (i), DMSP is first cleaved to form DMS and acrylate, and subsequently CoA is ligated with acrylate (*Figure 3—figure supplement 2A*). In this case, the intermediate acrylate is produced. (ii), CoA is primarily ligated with DMSP to form DMSP-CoA. Then, DMSP-CoA is cleaved, producing DMS and acryloyl-CoA (*Figure 3—figure supplement 2B*). In this scenario, the intermediate DMSP-CoA is produced. To determine the catalytic process of DddX, we monitored the occurrence of acrylate and/or DMSP-CoA in the reaction system via LC-MS. While acrylate was not detectable in the reaction system, a small peak of DMSP-CoA emerged after a 2 min reaction (*Figure 3F*), indicating that DMSP-CoA is primarily formed in the catalytic reaction of DddX, which is then cleaved to generate DMS and acryloyl-CoA.

Knowing the DddX enzyme activity, we examined its in vitro properties. The DddX enzyme had an optimal temperature and pH of 40°C and 8.5, respectively (*Figure 3—figure supplement 3A and B*). The apparent $K_M$ of DddX for ATP and CoA was 2.5 mM (*Figure 3—figure supplement 3C*) and 0.4 mM (*Figure 3—figure supplement 3D*), respectively. DddX had an apparent $K_M$ value of 0.4 mM for DMSP (*Figure 3—figure supplement 3E*), which is lower than that of most other reported DMSP lyases and the DMSP demethylase DmdA (*Supplementary file 1c*). The $k_{cat}$ of DddX for DMSP was 0.7 $s^{-1}$, with an apparent $k_{cat}/K_M$ of 1.6 × 10³ $M^{-1}$ $s^{-1}$. The catalytic efficiency of DddX toward DMSP is higher than known DMSP lyases DddK, DddP, DddD, but lower than DddY and Alma1 (*Supplementary file 1c*).

Despite DddX belongs to the ACD superfamily, the amino acid identity between DddX and known ACD enzymes is relatively low, with the highest being 26 % between DddX and the *Giardia lamblia* ACS (*Sánchez et al., 2000*). The $k_{cat}/K_M$ value of DddX towards DMSP is lower than several reported ACS enzymes towards acetate (*Chan et al., 2011*; *You et al., 2017*). Because ACS enzymes were reported to have promiscuous activity toward different short chain fatty acids, such as acetate and propionate (*Patel and Walt, 1987*), we tested the substrate specificity of DddX. The recombinant DddX exhibited no activity towards acetate or propionate (*Figure 3—figure supplement 4*), and the presence of acetate or propionate had little effects on the enzymatic activity of DddX toward DMSP (*Figure 3—figure supplement 5*), indicating that DddX cannot utilize acetate or propionate as a substrate. Furthermore, we tested the ability of the strain D2 to grow with acetate or propionate as the sole carbon source. The wild-type strain D2 could use acetate or propionate as sole carbon source but deletion of *dddX* has little effect on the growth of strain D2 on these substrates (*Figure 3—figure supplement 6*), suggesting that *dddX* is unlikely to be involved in acetate and propionate catabolism. Together, these results indicate that DddX does not function as an acetate-CoA ligase.

## The crystal structure and the catalytic mechanism of DddX

To elucidate the structural basis of DddX catalysis, we solved the crystal structure of DddX in complex with ATP by the single-wavelength anomalous dispersion method using a selenomethionine derivative (Se-derivative) (*Supplementary file 1d*). Although there are four DddX monomers arranged as a tetramer in an asymmetric unit (*Figure 4—figure supplement 1A*), gel filtration analysis indicated that DddX maintains a dimer in solution (*Figure 4—figure supplement 1B*). Each DddX monomer contains a CoA-binding domain and an ATP-grasp domain (*Figure 4A*), with one loop (Gly280-Tyr300) of the CoA-binding domain inserting into the ATP-grasp domain. ATP is bound in DddX mainly via hydrophilic interactions, including hydrogen bonds and salt bridges (*Figure 4B*). The overall structure of DddX is similar to that of NDP-forming acetyl-CoA synthetase ACD1 (*Weiße et al., 2016*; *Figure 4—figure supplement 2*), with a root mean square deviation (RMSD) between these two structures of

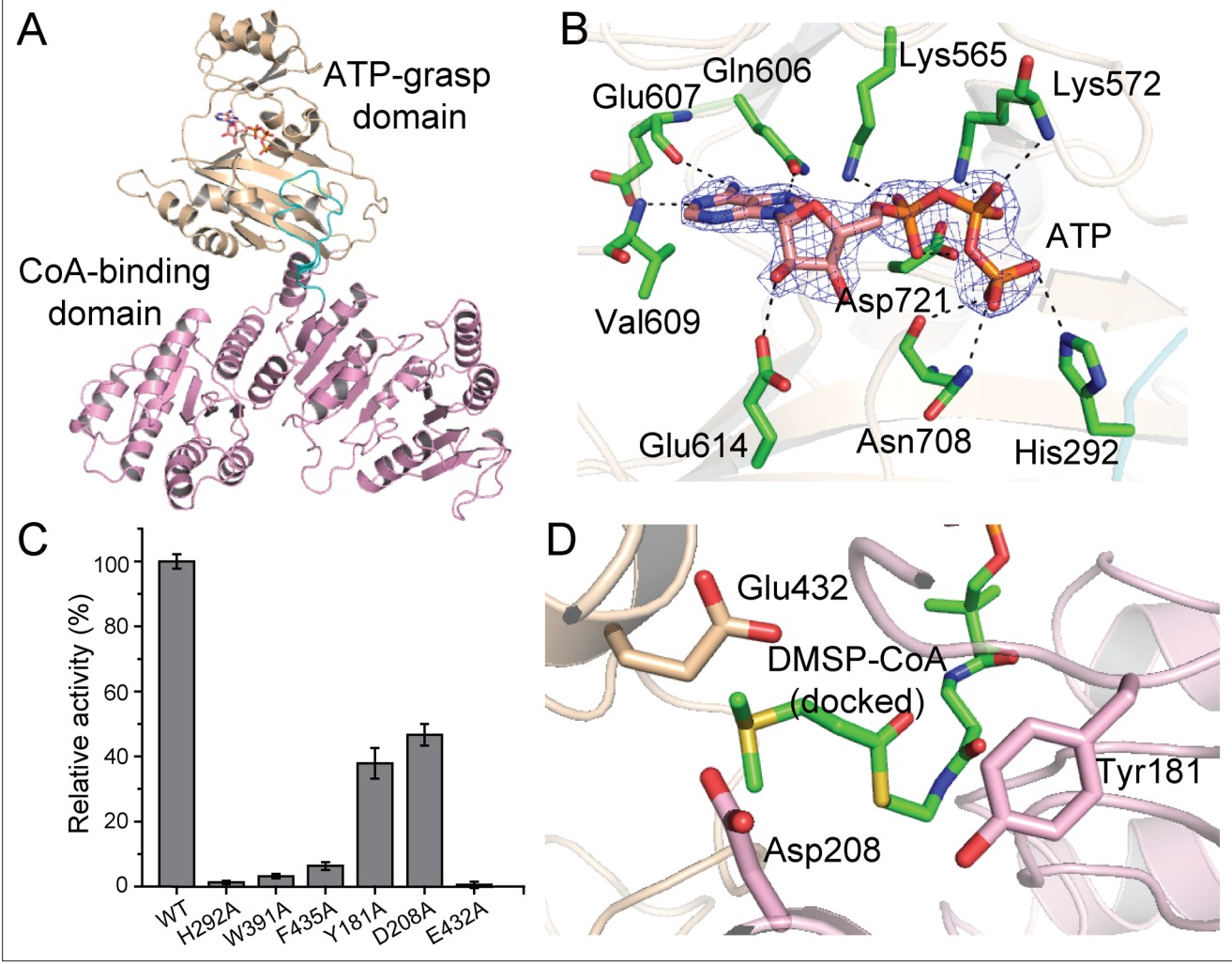

**Figure 4.** Structural and mutational analyses of DddX. (**A**) The overall structure of the DddX monomer. The DddX molecule contains a CoA-binding domain (colored in pink) and an ATP-grasp domain (colored in wheat). The loop region from the CoA-binding domain inserting into the ATP-grasp domain is colored in cyan. The ATP molecule is shown as sticks. (**B**) Residues of DddX involved in binding ATP. The $2F_o$ - $F_c$ densities for ATP are contoured in blue at 2.0σ. Residues of DddX involved in binding ATP are colored in green. (**C**) Enzymatic activities of DddX and its mutants. The activity of WT DddX was taken as 100%. (**D**) Structural analysis of the possible catalytic residues for the cleavage of DMSP-CoA. The docked DMSP-CoA molecule and the probable catalytic residues of DddX are shown as sticks.

The online version of this article includes the following figure supplement(s) for figure 4:

**Source data 1.** Enzymatic activities of DddX and its mutants.

**Figure supplement 1.** Structural and gel filtration analysis of DddX state of aggregation.

**Figure supplement 1—source data 1.** Gel filtration analysis of DddX.

**Figure supplement 2.** The overall structure of ACD1.

**Figure supplement 3.** Sequence alignment of DddX homologs, acetyl-CoA synthetases (ACS), and ATP-citrate lyases (ACLY).

**Figure supplement 4.** CD spectra of WT DddX and its mutants.

**Figure supplement 4—source data 1.** CD spectra of WT DddX and its mutants.

**Figure supplement 5.** Structural analysis of DddX docked with DMSP and CoA, and DMSP-CoA.

4.6 Å over 581 $C_\alpha$ atoms. ACD1 consists of separate α- and β-subunits (*Weiße et al., 2016*), which corresponds to the CoA-binding domain and the ATP-grasp domain of DddX, respectively.

Both DddX and ACS belong to the ACD superfamily, which also contains the well-studied ATP citrate lyases (ACLY) (*Weiße et al., 2016*; *Verschueren et al., 2019*; *Hu et al., 2017*). The biochemistry of DddX catalysis is similar to that of ACLY, which converts citrate to acetyl-CoA and oxaloacetate with

ATP and CoA as co-substrates (***Verschueren et al., 2019***; ***Hu et al., 2017***). The catalytic processes of enzymes in the ACD superfamily involve a conformational change of a 'swinging loop' or 'phosphohistidine segment', in which a conserved histidine is phosphorylated (***Weiße et al., 2016***; ***Verschueren et al., 2019***; ***Hu et al., 2017***). Sequence alignment indicated that His292 of DddX is likely the conserved histidine residue to be phosphorylated, and Gly280-Tyr300 is likely the 'swinging loop' (***Figure 4—figure supplement 3***). In the crystal structure of DddX, His292 from loop Gly280-Tyr300 directly forms a hydrogen bond with the γ-phosphate of ATP (***Figure 4B***), suggesting a potential for phosphorylation, which is further supported by mutational analysis. Mutation of His292 to alanine abolished the activity of DddX (***Figure 4C***), indicating the key role of His292 during catalysis. Circular-dichroism (CD) spectroscopy analysis showed that the secondary structure of His292Ala exhibits little deviation from that of wild-type (WT) DddX (***Figure 4—figure supplement 4***), indicating that the enzymatic activity loss was caused by amino acid replacement rather than by structural change. Altogether, these data suggest that His292 is phosphorylated in the catalysis of DddX on DMSP.

Having solved the crystal structure of the DddX-ATP complex, we next sought to determine the crystal structures of DddX in complex with CoA and DMSP. However, the diffractions of these crystals were poor and all attempts to solve the structures failed. Thus, we docked DMSP and CoA into the structure of DddX. In the docked structure, the CoA molecule is bound in the CoA-binding domain, while the DMSP molecule is bound in the interface between two DddX monomers (***Figure 4—figure supplement 5A***). Because our biochemical results demonstrated that DMSP-CoA is an intermediate of DddX catalysis (***Figure 3F***), we further docked DMSP-CoA into DddX. DMSP-CoA also locates between two DddX monomers (***Figure 4—figure supplement 5B***), and two aromatic residues (Trp391 and Phe435) form cation-π interactions with the sulfonium group of DMSP-CoA (***Figure 4—figure supplement 5C***). Mutations of these two residues significantly decreased the enzymatic activities of DddX (***Figure 4C***), suggesting that these residues play important roles in DddX catalysis. To cleave DMSP-CoA into DMS and acryloyl-CoA, a catalytic base is necessary to deprotonate DMSP-CoA. Structure analysis showed that Tyr181, Asp208, and Glu432 are close to the DMSP moiety (***Figure 4D***) and may function as the general base. Mutational analysis showed that the mutation of Glu432 to

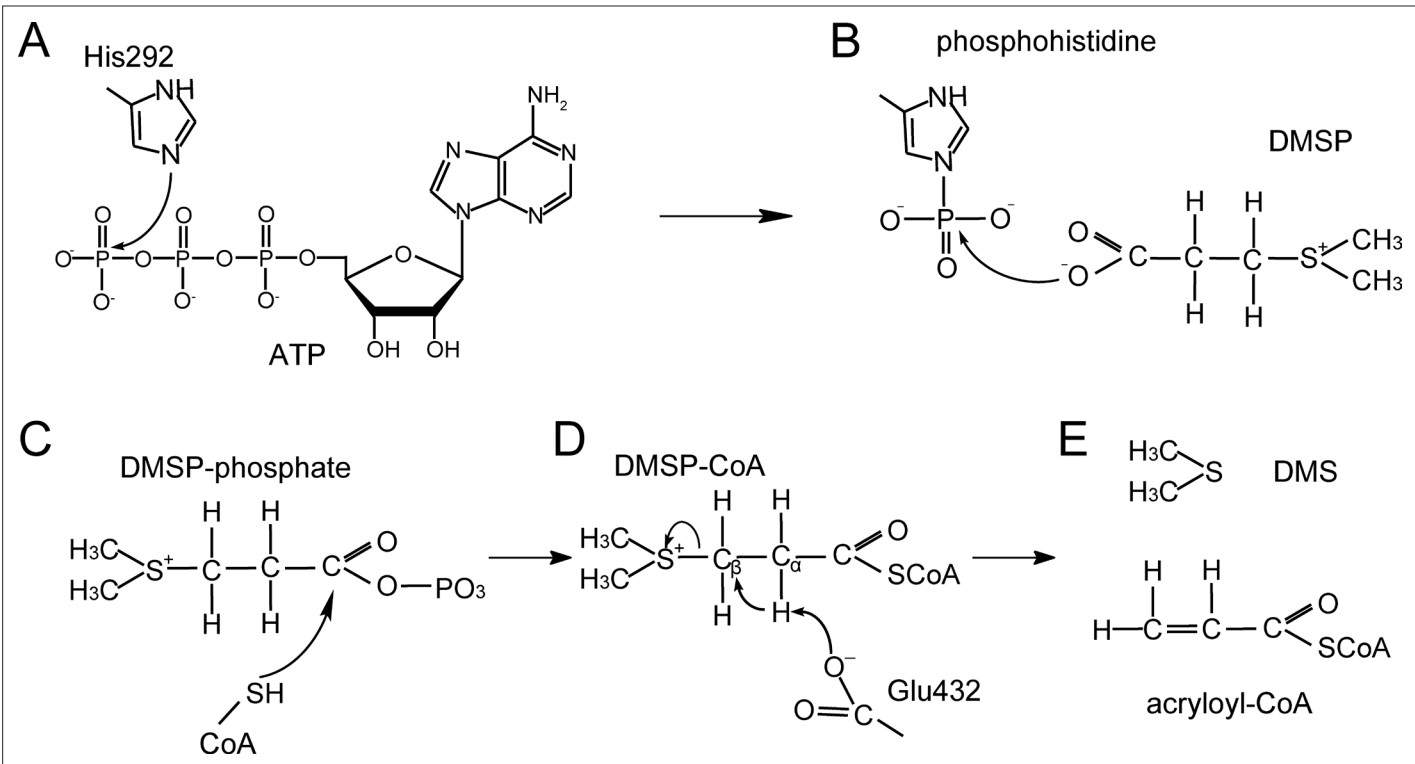

**Figure 5.** A proposed mechanism for DMSP cleavage to generate DMS and acryloyl-CoA catalyzed by DddX. (**A**) The residue His292 attacks the γ-phosphate of ATP. (**B**), The phosphoryl group is transferred from phosphohistidine to the DMSP molecule. (**C**), DMSP-phosphate is attacked by CoA. (**D**), The residue Glu432 acts as a general base to attack DMSP-CoA. (**E**), DMS and acryloyl-CoA are generated.

alanine abolished the enzymatic activity of DddX, while mutants Tyr181Ala and Asp208Ala still maintained ~40% activities (*Figure 4C*), indicating that Glu432 is the most probable catalytic residue for the final cleavage of DMSP-CoA. CD spectra of these mutants were indistinguishable from that of WT DddX (*Figure 4—figure supplement 4*), suggesting that the decrease in the enzymatic activities of the mutants were caused by residue replacement rather than structural alteration of the enzyme.

Based on structural and mutational analyses of DddX, and the reported molecular mechanisms of the ACD superfamily (*Weiße et al., 2016*; *Verschueren et al., 2019*; *Hu et al., 2017*), we proposed the molecular mechanism of DddX catalysis on DMSP (*Figure 5*). Firstly, His292 is phosphorylated by ATP, forming phosphohistidine (*Figure 5A*), which will be brought to the CoA-binding domain through the conformational change of the swinging loop Gly280-Tyr300. Next, the phosphoryl group is most likely transferred to DMSP to generate DMSP-phosphate (*Figure 5B*), which is subsequently attacked by CoA to form DMSP-CoA intermediate (*Figure 5C*). The last step is the cleavage of DMSP-CoA

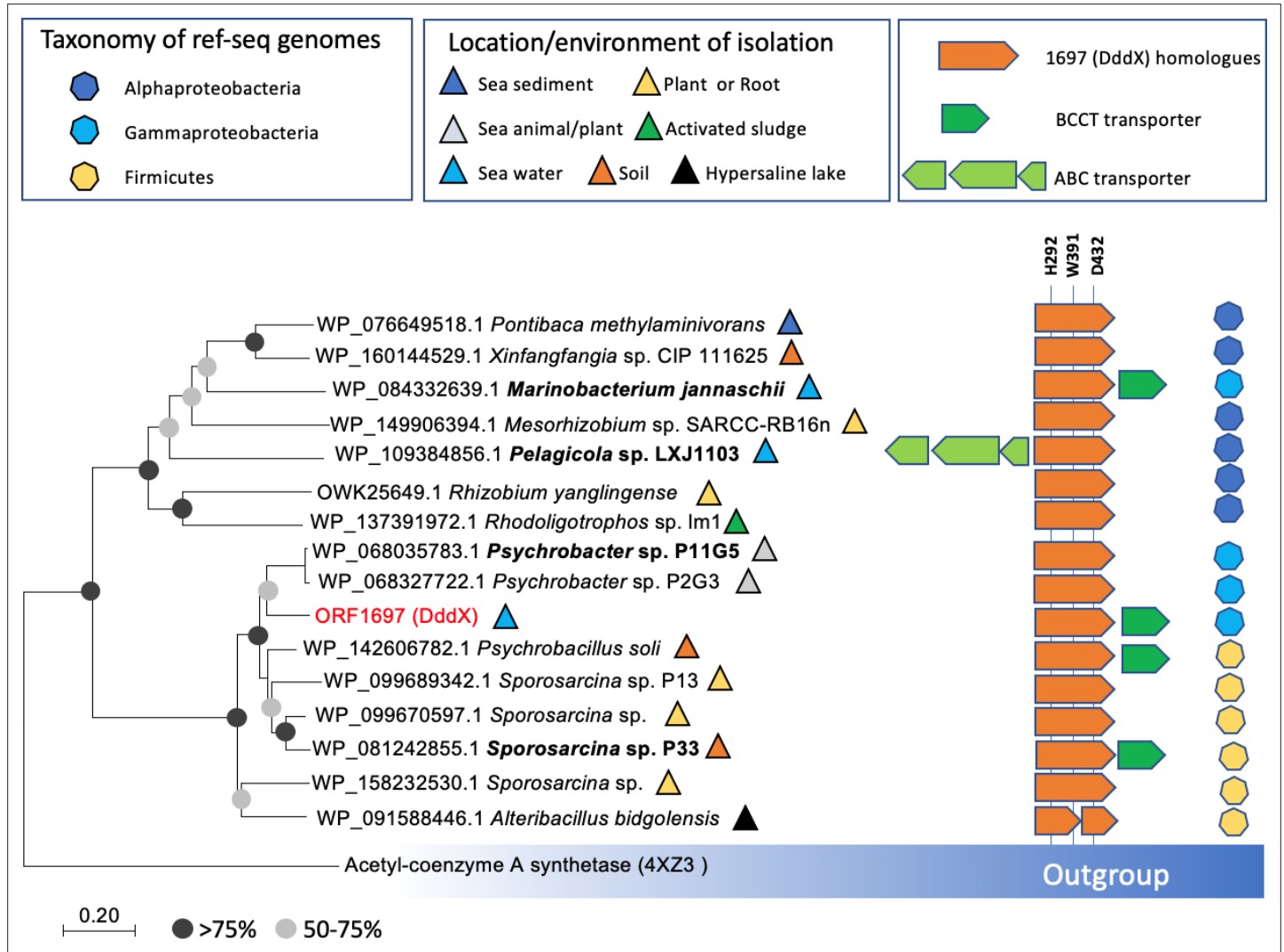

**Figure 6.** Distribution of DddX in bacterial genomes. The phylogenetic tree was constructed using neighbor-joining method in MEGA7. The acetyl-coenzyme A synthetase (ACS) (*Weiße et al., 2016*) was used as the outgroup. Sequence alignment was inspected for the presence of the key histidine residue (His292) involved in histidine phosphorylation that is known to be important for enzyme activity. A conserved Tyr391 is also found which is involved in cation-pi interaction with DMSP. The BCCT-type or ABC-type transporters for betaine-carnitine-choline-DMSP were found in the neighborhood of DddX in several genomes. Those DddX homologs that are functionally characterized (*Figure 6—figure supplement 1*) are highlighted in bold.

The online version of this article includes the following figure supplement(s) for figure 6:

**Figure supplement 1.** HPLC assay of the enzymatic activity of DddX homologs on DMSP at 260 nm.

probably initiated by the base-catalyzed deprotonation of Glu432 (*Figure 5D*). Finally, acryloyl-CoA and DMS are generated (*Figure 5E*) and released from the catalytic pocket of DddX.

## Distribution of DddX in bacteria

We next set out to determine the diversity and distribution of DddX in bacteria with sequenced genomes. We searched the NCBI Reference Sequence Database using the DddX sequence of *Psychrobacter* sp. D2 as the query. The data presented in *Figure 6* showed that DddX homologs are present in several diverse groups of bacteria, including Alphaproteobacteria, Gammaproteobacteria, and Firmicutes. Multiple sequence alignment showed the presence of the key residues involved in phosphorylation (H292), co-ordination of the substrate (e.g. W391) and catalysis (D432), suggesting that these DddX homologs are likely functional in bacterial DMSP catabolism. To further validate that these DddX homologs are indeed functional DMSP degrading enzymes, we chemically synthesized representative *dddX* sequences from Alphaproteobacteria (*Pelagicola* sp. LXJ1103), Gammaproteobacteria (*Psychrobacter* sp. P11G5; *Marinobacterium jannaschii*), and Firmicutes (*Sporosarcina* sp. P33). These candidate DddX enzymes were purified and all were shown to degrade DMSP and produce acryloyl-CoA confirming their predicted activity (*Figure 6—figure supplement 1*). We predict that bacteria containing DddX will have DMSP lyase activity, but this will depend on the expression of this enzyme in the host and substrate availability.

## Discussion

The cleavage of DMSP to produce DMS is a globally important biogeochemical reaction. Although all known DMSP lyases liberate DMS, they belong to different families, and likely evolved independently (*Bullock et al., 2017*). DddD belongs to the type III acyl CoA transferase family (*Todd et al., 2007*), DddP to the M24 metallopeptidase family enzyme (*Todd et al., 2009*), DddL/Q/W/K/Y to the cupin superfamily enzymes (*Lei et al., 2018*; *Li et al., 2017*) and Alma1 to the aspartate racemase superfamily (*Alcolombri et al., 2015*). To the best of our knowledge, DddX represents the first DMSP lyase of the ACD superfamily.

Of the reported DMSP lyases, only DddD catalyzes a two-step reaction which comprises a CoA transfer reaction and a cleavage reaction (*Alcolombri et al., 2014*). It is deduced that DMSP-CoA will be generated in the catalytic process of DddD (*Alcolombri et al., 2014*; *Curson et al., 2011b*; *Todd et al., 2007*). Despite this similarity, DddX is fundamentally different to DddD. Firstly, the co-substrates of DddX and DddD are different. ATP and CoA are essential co-substrates for the enzymatic activity of DddX, while for DddD catalysis, acetyl-CoA is used as a CoA donor, and ATP is not required (*Johnston et al., 2016*; *Alcolombri et al., 2014*). When CoA was replaced by acetyl-CoA in the reaction system, DddX failed to catalyze the cleavage of DMSP (*Figure 1—figure supplement 1*). Secondly, the products of DddD and DddX are different. DddD converts DMSP to DMS and 3-HP-CoA, whereas DddX produces DMS and acryloyl-CoA from DMSP. Except for DddD and DddX, all the other DMSP lyases cleave DMSP to DMS and acrylate.

It has been reported that accumulation of acryloyl-CoA is toxic to bacteria (*Reisch et al., 2013*; *Wang et al., 2017*; *Cao et al., 2017*; *Todd et al., 2012*). Thus, *Psychrobacter* sp. D2 requires an efficient system to metabolize the acryloyl-CoA produced from DMSP lysis by DddX. With the transcription of genes *1,698* and *1699,*, directly downstream of *dddX* and likely co-transcribed with *dddX*, being significantly enhanced by growth on DMSP, their enzyme products (DddC and DddB) likely participate in the metabolism and detoxification of acryloyl-CoA or downstream metabolites. However, the recombinant 1698 and However, 1699 exhibited no enzymatic activity on acryloyl-CoA.

The *Psychrobacter* sp. D2 genome also contains *acuI* and *acuH* homologs (*2674*, *0105*, *1810*, *1,692*, and *1695*) (*Supplementary file 1e*), which may directly act on acryloyl-CoA to produce propionate-CoA or 3-HP-CoA (*Reisch et al., 2013*; *Wang et al., 2017*; *Cao et al., 2017*; *Todd et al., 2012*). If *Psychrobacter* sp. D2 employs its AcuH homolog to convert acryloyl-CoA to 3-HP-CoA (*Cao et al., 2017*), then, given the high-sequence identity of 1,698 to DddC and 1699 to DddB, it is possible that these enzymes further catabolize 3-HP-CoA to acetyl-CoA (*Alcolombri et al., 2014*; *Curson et al., 2011b*). Furthermore, we showed that the recombinant 0105, an AcuI homolog, could act on acryloyl-CoA to produce propionate-CoA with NADPH as a cofactor (*Figure 1—figure supplement 2*). Thus, *Psychrobacter* sp. D2 may also employ an AcuI (*i.e*. 0105) to convert acryloyl-CoA to

propionate-CoA (*Figure 1*), which would be metabolized through the methylmalonyl-CoA pathway (*Reisch et al., 2013*).

Several DMSP catabolizing bacteria, e.g. *Halomonas* HTNK1 with DddD, are reported to utilize acrylate as the carbon source for growth via e.g. *acuN, acuK, acul, acuH,* and *prpE* gene products (*Curson et al., 2011a*; *Reisch et al., 2013*; *Todd et al., 2010*). Despite the presence of several *acuN, acuK, acul, acuH* and *prpE* homologs in its genome (*Supplementary file 1e*), *Psychrobacter* sp. D2 could not use acrylate as a sole carbon source (*Figure 2A*). Thus, *Psychrobacter* sp. D2 either (i), lacks a functional acrylate transporter; (ii), these homologs that are predicted to be involved in acrylate metabolism are not functional in vivo; or (iii), these genes are not induced by acrylate. Clearly further biochemical and genetic experiments are required to establish the how acryloyl-CoA is catabolized in this bacterium.

Many marine bacteria, especially roseobacters, are reported to metabolize DMSP via more than one pathway (*Curson et al., 2011b*; *Bullock et al., 2017*). For example, *Ruegeria pomeroyi* DSS-3, one of the type strains of the marine *Roseobacter* clade, possesses both the demethylation and the lysis pathway for DMSP metabolism (*Reisch et al., 2013*). Moreover, it contains multiple *ddd* genes (*dddQ, dddP* and *dddW*) (*Reisch et al., 2013*; *Todd et al., 2011*). DmdA homologs were not identified in the genome of *Psychrobacter* sp. D2, indicating that the demethylation pathway is absent in strain D2. The fact that the mutant Δ*dddX* could not produce DMS from DMSP and was unable to grow on DMSP as the sole carbon source suggests that *Psychrobacter* sp. D2 only possesses one DMSP lysis pathway for DMSP degradation. Why some bacteria have evolved multiple DMSP utilization pathways and some bacteria only possess one pathway awaits further investigation.

Here, we demonstrate that DddX is a functional DMSP lyase present in several isolates of Gammaproteobacteria, Alphaproteobacteria and, notably, Gram-positive Firmicutes, for example in *Sporosarcina* sp. P33. The distribution of DddX in these bacterial lineages points to the role of horizontal gene transfer (HGT) in the dissemination of *dddX* in environmental bacteria and this certainly warrants further investigation. Interestingly, DddX is found in several bacterial isolates which were isolated from soil or plant roots, suggesting that DMSP may also be produced in these ecosystems. Finally, it has been reported that many other Gram-positive actinobacteria can make DMS from DMSP (*Liu et al., 2018*). Interestingly, these Actinobacteria lack *dddX* and any other known DMSP lyase genes. Thus, there is still more biodviversity in microbial DMSP lyases to be uncovered.

## Conclusion

DMSP is widespread in nature and cleavage of DMSP produces DMS, an important mediator in the global sulfur cycle. In this study, we report the identification of a novel ATP-dependent DMSP lyase DddX from marine bacteria. DddX belongs to the ACD superfamily, and catalyzes the conversion of DMSP to DMS and acryloyl-CoA, with CoA and ATP as co-substrates. DddX homologs are found in both Gram-positive and Gram-negative bacterial lineages. This study offers new insights into how diverse bacteria cleave DMSP to generate the climatically important gas DMS.

## Materials and methods

**Key resources table**

| Reagent type (species) or resource | Designation | Source or reference | Identifiers | Additional information |
|---|---|---|---|---|
| Strain, strain background (*Psychrobacter* sp.) | D2 | This study; Zhang Laboratory | | Wild-type isolate; Available from Zhang lab |
| Strain, strain background (*Psychrobacter* sp.) | Δ*dddX* | This study; Zhang Laboratory | | the *dddX* gene deletion mutant of *Psychrobacter* sp. D2; Available from Zhang lab |
| Strain, strain background (*Psychrobacter* sp.) | Δ*dddX*/pBBR1MCS-*dddX* | This study; Zhang Laboratory | | Δ*dddX* containing pBBR1MCS-*dddX* plasmid; Available from Zhang lab |

*Continued on next page*

*Continued*

| Reagent type (species) or resource | Designation | Source or reference | Identifiers | Additional information |
|---|---|---|---|---|
| Strain, strain background (*Psychrobacter* sp.) | Δ*dddX*/pBBR1MCS | This study; Zhang Laboratory | | Δ*dddX* containing pBBR1MCS plasmid; Available from Zhang lab |
| Strain, strain background (*Escherichia coli*) | WM3064 | **Dehio and Meyer, 1997** | | Conjugation donor strain |
| Strain, strain background (*Escherichia coli*) | DH5α | Vazyme Biotech company (China) | | Transformed cells for gene cloning |
| Strain, strain background (*Escherichia coli*) | BL21(DE3) | Vazyme Biotech company (China) | | Transformed cells for gene expression |
| Recombinant DNA reagent | pK18*mobsacB*-Ery | **Wang et al., 2015b** | | Gene knockout vector |
| Recombinant DNA reagent | pK18Ery-*dddX* | This study; Zhang Laboratory | | pK18*mobsacB*-Ery containing the homologous arms of the *dddX* gene of *Psychrobacter*. sp. D2; Available from Zhang lab |
| Recombinant DNA reagent | pBBR1MCS | **Kovach et al., 1995** | | Broad-host-range cloning vector |
| Recombinant DNA reagent | pBBR1MCS-*dddX* | This study; Zhang Laboratory | | pBBR1MCS containing the *dddX* gene and its promoter of *Psychrobacter*. sp. D2; Available from Zhang lab |
| Recombinant DNA reagent | pET-22b-*dddX* | This study; Zhang Laboratory | | Used for *dddX* expression; Available from Zhang lab |
| Commercial assay or kit | Pierce BCA Protein Assay Kit | Thermo, USA | | Protein assay |
| Commercial assay or kit | Bacterial genomic DNA isolation kit | BioTeke Corporation, China | | DNA extraction |
| Commercial assay or kit | RNeasy Mini Kit | QIAGEN, America | | RNA extraction |
| Commercial assay or kit | PrimeScript RT reagent Kit | Takara, Japan | | Reverse transcription |
| Commercial assay or kit | Genome sequencing of *Psychrobacter* sp. D2 | Biozeron Biotechnology Co., Ltd, China | NCBI: JACDXZ000000000 | |
| Commercial assay or kit | Transcriptome sequencing of *Psychrobacter* sp. D2 | BGI Tech Solutions Co., Ltd, China | NCBI: PRJNA646786 | |
| Software, algorithm | HKL3000 program | **Minor et al., 2006** | | Diffraction data analysis |
| Software, algorithm | CCP4 program Phaser | **Winn et al., 2011** | | Diffraction data analysis |
| Software, algorithm | Coot | **Emsley et al., 2010** | | Diffraction data analysis |
| Software, algorithm | *Phenix* | **Adams et al., 2010** | | Diffraction data analysis |
| Software, algorithm | PyMOL | Schrödinger, LLC | | http://www.pymol.org/ |
| Software, algorithm | MEGA 7 | **Kumar et al., 2016** | | Phylogenetic analysis |

## Bacterial strains, plasmids, and growth conditions

Strains and plasmids used in this study are shown in *Supplementary file 1f*. Isolates were cultured in the marine broth 2,216 medium or the basal medium (*Supplementary file 1g*) with 5 mM DMSP as the sole carbon source at 15°C–25°C. *Psychrobacter* sp. D2 was cultured in the marine broth 2,216 medium or the basal medium (*Supplementary file 1g*) supplied with different carbon sources (sodium pyruvate, acrylate or DMSP at a final concentration of 5 mM) at 15–25°C. The *E. coli* strains DH5α and BL21(DE3) were grown in the Lysogeny Broth (LB) medium at 37°C. Diaminopimelic acid (0.3 mM) was added to culture the *E. coli* WM3064 strain.

## Isolation of bacterial strains from Antarctic samples

A total of five samples were collected from the Great Wall Station of Antarctica during the Chinese Antarctic Great Wall Station Expedition in January, 2017. Information of samples is shown in *Figure 2—figure supplement 1* and *Supplementary file 1a*. Algae and sediments were collected using a grab sampler and stored in airtight sterile plastic bags at 4°C. Seawater samples were filtered through polycarbonate membranes with 0.22 μm pores (Millipore Co., United States). The filtered membranes were stored in sterile tubes (Corning Inc, United States) at 4°C. All samples were transferred into a 50 ml flask containing 20 ml 3% (w/v) seasalt solution (SS) and shaken at 100 rpm at 15°C for 2 hr. The suspension obtained was subsequently diluted to $10^{-6}$ with sterile SS. An aliquot (200 μl) of each dilution was spread on the basal medium (*Supplementary file 1g*) plates with 5 mM DMSP as the sole carbon source. The plates were then incubated at 15°C in the dark for 2–3 weeks. Colonies with different appearances were picked up and were further purified by streaking on the marine 2,216 agar plates for at least three passages. The abilities of the colonies for DMSP catabolism were verified in a liquid basal medium with DMSP (5 mM) as the sole carbon source. The isolates were stored at –80°C in the marine broth 2,216 medium containing 20 % (v/v) glycerol.

## Sequence analysis of bacterial 16s rRNA genes

Genomic DNA of the isolates was extracted using a bacterial genomic DNA isolation kit (BioTeke Corporation, China) according to the manufacturer's instructions. The 16 S rRNA genes of these strains were amplified using the primers 27 F/1492 R (*Supplementary file 1h*) and sequenced to determine their taxonomy. Pairwise similarity values for the 16 S rRNA gene of the cultivated strains were calculated through the EzBiocloud server (http://www.ezbiocloud.net/) (*Yoon et al., 2017*).

## Bacterial growth assay with DMSP as the sole carbon source

Cells were grown in the marine broth 2,216 medium, harvested after incubation at 15 °C for 24 hr, and then washed three times with sterile SS. The washed cells were diluted to the same density of $OD_{600} \approx 2.0$, and then 1 % (v/v) cells were inoculated into the basal medium with DMSP, sodium acetate, or sodium propionate (5 mM) as the sole carbon source. The bacteria were cultured in the dark at 15°C. The growth of the bacteria was measured by detecting the $OD_{600}$ of the cultures at different time points using a spectrophotometer V-550 (Jasco Corporation, Japan).

## Quantification of DMS by GC

To measure the production of DMS, cells were first cultured overnight in the marine broth 2,216 medium, and then washed three times with sterile SS. The washed cells were diluted to the same density of $OD_{600} \approx 0.3$, then diluted 1:10 into vials (Anpel, China) containing the basal medium supplied with 5 mM DMSP as the sole carbon source. The vials were crimp sealed with rubber bungs and incubated for 2 hr at 25°C. The cultures were then assayed for DMS production on a gas chromatograph (GC-2030, Shimadzu, Japan) equipped with a flame photometric detector (*Liu et al., 2018*). An eight-point calibration curve of DMS standards was used (*Curson et al., 2017*). Abiotic controls of the basal medium amended with 5 mM DMSP were set up and incubated under the same conditions to monitor the background lysis of DMSP to DMS. Following growth of all bacteria strains in the marine broth 2,216 medium, cells were collected by centrifugation, resuspended in the lysis buffer (50 mM Tris-HCl, 100 mM NaCl, 0.5% glycerol, pH 8.0), and lysed by sonicated. The protein content in the cells was measured by Pierce BCA Protein Assay Kit (Thermo, USA). DMS production is expressed as nmol $min^{-1}$ mg $protein^{-1}$.

## Transcriptome sequencing of *Psychrobacter* sp. D2

Cells of strain D2 were cultured in the marine broth 2,216 medium at 180 rpm at 15°C for 24 hr. The cells were collected and washed three times with sterile SS, and then cultured in sterile SS at 180 rpm at 15°C for 24 hr. Subsequently, the cells were washed twice with sterile SS, and incubated at 4°C for 24 hr. After incubation, the cells were harvested and resuspended in sterile SS, which were used as the resting cells. The resting cells were inoculated into the basal medium with DMSP (5 mM) as the sole carbon source, and incubated at 180 rpm at 15°C. When the $OD_{600}$ of the cultures reached 0.3, the cells were harvested. The resting cells and those cultured in the basal medium with sodium pyruvate (5 mM) as the sole carbon source were set up as controls. Total RNA was extracted using

a RNeasy Mini Kit (QIAGEN, America) according to the manufacturer's protocol. After validating the quality, RNA samples were sent to BGI Tech Solutions Co., Ltd (China) for transcriptome sequencing and subsequent bioinformatic analysis.

## Real-time qPCR analysis

Cells of *Psychrobacter* sp. D2 were cultured in the marine broth 2,216 medium at 180 rpm at 15°C to an $OD_{600}$ of 0.8. Then, cells were induced by 5 mM DMSP, and the control group without DMSP was also set up. After 20 min's induction, total RNA was extracted using a RNeasy Mini Kit (Qiagen, Germany) according to the manufacturer's instructions. Genomic DNA was removed using gDNA Eraser (TaKaRa, Japan) and cDNA was synthesized using a PrimeScript RT reagent Kit. The qPCR was performed on the Light Cycler II 480 System (Roche, Switzerland) using a SYBR Premix Ex Taq (TaKaRa, Japan). Relative expression levels of target genes were calculated using the LightCycler480 software with the 'Advanced Relative Quantification' method. The *recA* gene was used as an internal reference gene. The primers used in this study are shown in *Supplementary file 1h*.

## Genetic manipulations of *Psychrobacter* sp. D2

Deletion of the *dddX* gene was performed via pK18*mobsacB*-Ery-based homologue recombination (*Wang et al., 2015a*). The upstream and downstream homologous sequences of the *dddX* gene were amplified with primer sets *dddX*-UP-F/*dddX*-UP-R and *dddX*-Down-F/*dddX*-Down-R, respectively. Next, the PCR fragments were inserted to the vector pK18*mobsacB*-Ery with *Hind*III/*BamH*I as the restriction sites to generate pK18Ery-*dddX*, which was transferred into *E. coli* WM3064. The plasmid pK18Ery-*dddX* was then mobilized into *Psychrobacter* sp. D2 by intergeneric conjugation with *E. coli* WM3064. To select for colonies in which the pK18Ery-*dddX* had integrated into the *Psychrobacter* sp. D2 genome by a single crossover event, cells were plated on the marine 2,216 agar plates containing erythromycin (25 µg/ml). Subsequently, the resultant mutant was cultured in the marine broth 2,216 medium and plated on the marine 2,216 agar plates containing 10% (w/v) sucrose to select for colonies in which the second recombination event occurred. Single colonies appeared on the plates were streaked on the marine 2,216 agar plates containing erythromycin (25 µg/ml), and colonies sensitive to erythromycin were further validated to be the *dddX* gene deletion mutants by PCR with primer pairs of *dddX*-1000-F/*dddX*-1000-R and *dddX*-300Up-F/*dddX*-700Down-R.

For complementation of the Δ*dddX* mutant, the *dddX* gene with its native promoter was amplified using the primers set *dddX*-pBBR1-PF/*dddX*-pBBR1-PR. The PCR fragment was digested with *Kpn*I and *Xho*I, and then inserted into the vector pBBR1MCS to generate pBBR1MCS-*dddX*. This plasmid was then transformed into *E. coli* WM3064, and mobilized into the Δ*dddX* mutant by intergeneric conjugation. After mating, the cells were plated on the marine 2,216 agar plates containing kanamycin (80 µg/ml) to select for the complemented mutant. The empty vector pBBR1MCS was mobilized into the Δ*dddX* mutant using the same protocol. Colony PCR was used to confirm the presence of the transferred plasmid. The strains, plasmids and primers used in this study are shown in *Supplementary file 1f and h*.

## Gene cloning, point mutation, and protein expression and purification

The 2247 bp full-length *dddX* gene was amplified from the genome of *Psychrobacter* sp. D2 by PCR using *FastPfu* DNA polymerase (TransGen Biotech, China). The amplified gene was then inserted to the *Nde*I/*Xho*I restriction sites of the pET-22b vector (Novagen, Germany) with a C-terminal His tag. All the point mutations in DddX were introduced using the PCR-based method and verified by DNA sequencing. The DddX protein and its mutants were expressed in *E. coli* BL21 (DE3). The cells were cultured in the LB medium with 0.1 mg/ml ampicillin at 37 °C to an $OD_{600}$ of 0.8–1.0 and then induced at 18°C for 16 hr with 0.5 mM isopropyl-β-D-thiogalactopyranoside (IPTG). After induction, cells were collected by centrifugation, resuspended in the lysis buffer (50 mM Tris-HCl, 100 mM NaCl, 0.5% glycerol, pH 8.0), and lysed by pressure crusher. The proteins were first purified by affinity chromatography on a $Ni^{2+}$-NTA column (GE healthcare, America), and then fractionated by anion exchange chromatography on a Source 15Q column (GE healthcare, America) and gel filtration on a Superdex G200 column (GE healthcare, America). The Se-derivative of DddX was overexpressed in *E. coli* BL21 (DE3) under 0.5 mM IPTG induction in the M9 minimal medium supplemented with selenomethionine,

lysine, valine, threonine, leucine, isoleucine, and phenylalanine. The recombinant Se-derivative was purified using the aforementioned protocol for the wild-type DddX.

## Enzyme assay and product identification

For the routine enzymatic activity assay of the DddX protein, the purified DddX protein (at a final concentration of 0.1 mM) was incubated with 1 mM DMSP, 1 mM CoA, 1 mM ATP, 2 mM MgCl$_2$ and 100 mM Tris-HCl (pH 8.0). The reaction was performed at 37°C for 0.5 hr, and terminated by adding 10% (v/v) hydrochloric acid. The control groups had the same reaction system except that the DddX protein was not added. DMS was detected by GC as described above. Products of acryloyl-CoA and DMSP-CoA were analyzed using LC-MS. Components of the reaction system were separated on a reversed-phase SunFire C$_{18}$ column (Waters, Ireland) connected to a high performance liquid chromatography (HPLC) system (Dionex, United States). The ultraviolet absorbance of samples was detected by HPLC under 260 nm. The samples were eluted with a linear gradient of 1–20% (v/v) acetonitrile in 50 mM ammonium acetate (pH 5.5) over 24 min. The HPLC system was coupled to an impact HD mass spectrometer (Bruker, Germany) for *m/z* determination. To determine the optimal temperature for DddX enzymatic activity, reaction mixtures containing 5 mM DMSP, 5 mM CoA, 5 mM ATP, 6 mM MgCl$_2$, 100 mM Tris-HCl (pH 8.5), and 10 µM DddX were incubated at 5–50°C (with a 5°C interval) for 15 min. The optimum pH for DddX enzymatic activity was examined at 40°C (the optimal temperature for DddX enzymatic activity) using Britton-Robinson Buffer (*Britton, 1952*) with pH from 7.5 to 11.0, with a 0.5 interval. The kinetic parameters of DddX were measured by determining the production of DMS with nonlinear analysis based on the initial rates, and all the measurements were performed at the optimal pH and temperature.

The enzymatic activity of DddX toward sodium acetate or sodium propionate was measured by determining the production of acetyl-CoA or propionyl-CoA using HPLC as described above with DMSP replaced by sodium acetate or sodium propionate. To determine the effects of sodium acetate or sodium propionate on the enzymatic activity of DddX toward DMSP, sodium acetate or sodium propionate at a final concentration of 1 mM, 2 mM or 5 mM were individually added to the reaction mixture. All the measurements were performed at the optimum pH and temperature for DddX.

The enzymatic activity of 0105 (AcuI) toward acryloyl-CoA was measured by determining the production of propionate-CoA using HPLC as described above. The reaction mixture contained 2 mM DMSP, 2 mM CoA, 2 mM ATP, 10 mM MgCl$_2$, 1 mM NADPH, 100 mM Tris-HCl (pH 8.5), 0.1 mM DddX, and 0.9 mM 0105. The reaction was performed at 40°C, pH 8.5 for 2 hr, and terminated by adding 10 % (v/v) hydrochloric acid.

## Crystallization and data collection

The purified DddX protein was concentrated to ~8 mg/ml in 10 mM Tris–HCl (pH 8.0) and 100 mM NaCl. The DddX protein was mixed with ATP (1 mM), and the mixtures were incubated at 0°C for 1 hr. Initial crystallization trials for DddX/ATP complex were performed at 18°C using the sitting-drop vapor diffusion method. Diffraction-quality crystals of DddX/ATP complex were obtained in hanging drops containing 0.1 M lithium sulfate monohydrate, 0.1 M sodium citrate tribasic dihydrate (pH 5.5) and 20 % (w/v) polyethylene glycol (PEG) 1000 at 18°C after 2-week incubation. Crystals of the DddX Se-derivative were obtained in hanging drops containing 0.1 M HEPES (pH 7.5), 10% PEG 6000% and 5% (v/v) (+/-)–2-Methyl-2,4-pentanediol at 18°C after 2-week incubation. X-ray diffraction data were collected on the BL18U1 and BL19U1 beamlines at the Shanghai Synchrotron Radiation Facility. The initial diffraction data sets were processed using the HKL3000 program with its default settings (*Minor et al., 2006*).

## Structure determination and refinement

The crystals of DddX/ATP complex belong to the C2 space group, and Se-derivative of DddX belong to the $P2_12_12_1$ space group. The structure of DddX Se-derivative was determined by single-wavelength anomalous dispersion phasing. The crystal structure of DddX/ATP complex was determined by molecular replacement using the CCP4 program Phaser (*Winn et al., 2011*) with the structure of DddX Se-derivative as the search model. The refinements of these structures were performed using Coot (*Emsley et al., 2010*) and *Phenix* (*Adams et al., 2010*). All structure figures were processed using the program PyMOL (http://www.pymol.org/).

## Circular dichroism (CD) spectroscopy

CD spectra for WT DddX and its mutants were carried out in a 0.1 cm-path length cell on a JASCO J-1500 Spectrometer (Japan). All proteins were adjusted to a final concentration of 0.2 mg/ml in 10 mM Tris-HCl (pH 8.0) and 100 mM NaCl. Spectra were recorded from 250 to 200 nm at a scan speed of 200 nm/min.

## Molecular docking simulations

The structure of the DddX/ATP complex containing a pair of subunits, α and β was loaded and energy minimised in Flare (v3.0, Cresset) involving 11,248 moving heavy atoms (Chain A: 5312, Chain B: 5312, Chain G: 10 and Chain S Water: 614). The molecule minimized with 2000 iterations using a gradient of 0.657 kcal/A. The minimised structure had an RMSD 0.82 Å relative to the starting structure and a decrease in starting energy from 134999.58 kcal/mol to a final energy of 6888.60 kcal/mol. The DMSP, CoA and DMSP-CoA molecules were drawn in MarvinSketch (v19.10.0, 2019, ChemAxon for Mac) and exported as a Mol SDF format. The molecules were imported into Flare and docked into the proposed CoA/DMSP binding site using the software's default docking parameters for intensive pose searching and scoring.

## Identification of DddX homologs in bacteria and phylogenetic analysis

DddX (1697) of *Psychrobacter* sp. D2 was used as the query sequence to search for homologs in genome-sequenced bacteria in the NCBI Reference Sequence Database (RefSeq, https://www.ncbi.nlm.nih.gov/refseq/) using BLastP with a stringent setting with an e-value cut-off < –75, sequence coverage >70% and percentage identity >30%. These high stringency settings are necessary to exclude other acetyl-CoA synthetase family proteins (ACS) which are unlikely to be involved in DMSP catabolism. Multiple sequence alignment was carried out using MEGA 7 (*Kumar et al., 2016*) and the presence of histidine 292, tryptophan 391 and glutamate 432 was manually inspected. To confirm the activity of DddX homologs from retrieved sequences from these genome-sequenced bacteria, four sequences (*Sporosarcina* sp. P33; *Psychrobacter* sp. P11G5; *Marinobacterium jannaschii; Pelagicola* sp. LXJ1103) were chemically synthesized and their enzyme activity for DMSP degradation was confirmed experimentally (*Figure 6—figure supplement 1*). The phylogenetic tree was constructed using the neighbour-joining method with 500 bootstraps using MEGA 7 (*Kumar et al., 2016*). The characterized ACS ACD1 (*Weiße et al., 2016*) was used as the outgroup.

## Acknowledgements

We thank the staffs from BL18U1 & BL19U1 beamlines of National Facility for Protein Sciences Shanghai (NFPS) and Shanghai Synchrotron Radiation Facility, for assistance during data collection. We thank Caiyun Sun and Jingyao Qu from State Key laboratory of Microbial Technology of Shandong University for their help in HPLC and LC-MS. This work was supported by the National Key Research and Development Program of China (2018YFC1406700, 2016YFA0601303), the National Science Foundation of China (grants 91851205, 31630012, U1706207, 42076229, 31870052, 31800107), Major Scientific and Technological Innovation Project (MSTIP) of Shandong Province (2019JZZY010817), the Program of Shandong for Taishan Scholars (tspd20181203) and Natural Environment Research Council Standard grants (NE/N002385, NE/P012671 and NE/S001352).

## Additional information

### Funding

| Funder | Grant reference number | Author |
| --- | --- | --- |
| National Key Research and Development Program of China | 2018YFC1406700 | Yu-Zhong Zhang |
| National Science Foundation of China | 91851205 | Yu-Zhong Zhang |

| Funder | Grant reference number | Author |
| --- | --- | --- |
| National Science Foundation of China | 31630012 | Yu-Zhong Zhang |
| National Key Research and Development Program of China | 2016YFA0601303 | Yu-Zhong Zhang |
| National Science Foundation of China | U1706207 | Yu-Zhong Zhang |
| National Science Foundation of China | 42076229 | Chun-Yang Li |
| National Science Foundation of China | 31870052 | Xiu-Lan Chen |
| National Science Foundation of China | 31800107 | Peng Wang |
| Major Scientific and Technological Innovation Project of Shandong Province | 2019JZZY010817 | Yu-Zhong Zhang |
| Program of Shandong for Taishan Scholars | tspd20181203 | Yu-Zhong Zhang |
| Natural Environment Research Council | NE/N002385 | Jonathan D Todd |
| Natural Environment Research Council | NE/P012671 | Jonathan D Todd |
| Natural Environment Research Council | NE/S001352 | Jonathan D Todd |

The funders had no role in study design, data collection and interpretation, or the decision to submit the work for publication.

## Author contributions

Chun-Yang Li, Investigation, Validation, Writing – original draft; Xiu-Juan Wang, Mussa Quareshy, Branko Rihtman, Investigation, Writing – original draft; Xiu-Lan Chen, supervision, Validation, writing-review-and-editing; Qi Sheng, Shan Zhang, Peng Wang, Xuan Shao, Chao Gao, Investigation; Fuchuan Li, Shengying Li, Weipeng Zhang, Xiao-Hua Zhang, Gui-Peng Yang, Jonathan D Todd, writing-review-and-editing; Yin Chen, Investigation, Writing – original draft, writing-review-and-editing; Yu-Zhong Zhang, conceptualization, funding-acquisition, supervision

## Author ORCIDs

Chun-Yang Li (ID) http://orcid.org/0000-0002-1151-4897
Yin Chen (ID) http://orcid.org/0000-0002-0367-4276
Yu-Zhong Zhang (ID) http://orcid.org/0000-0002-2017-1005

## Decision letter and Author response

Decision letter https://doi.org/10.7554/eLife.64045.sa1
Author response https://doi.org/10.7554/eLife.64045.sa2

# Additional files

## Supplementary files

• Supplementary file 1. Tables listing sampling information, homology alignment results, kinetic parameters, crystallographic data, bacterial strains, plasmids, medium composition, and primers used in this study. (**a**) Information of the Antarctic samples used in this study. (**b**) Homology alignment of proteins in *Psychrobacter* sp. D2 with known DMSP lyases. (**c**) Kinetic parameters of DMSP lyases and DMSP demethylase DmdA. (**d**) Crystallographic data collection and refinement parameters of DddX. (**e**) Homology alignment of proteins in *Psychrobacter* sp. D2 with known enzymes involved in acrylate catabolism. (**f**) Strains and plasmids used in this study. (**g**) Composition

of the basal medium (lacking the carbon source). (**h**) Primers used in this study.

• Transparent reporting form

### Data availability

The draft genome sequences of Psychrobacter sp. D2 have been deposited in the National Center for Biotechnology Information (NCBI) Genome database under accession number JACDXZ000000000. All the RNA-seq read data have been deposited in NCBI's sequence read archive (SRA) under project accession number PRJNA646786. The structure of DddX/ATP complex has been deposited in the PDB under the accession code 7CM9.

The following dataset was generated:

| Author(s) | Year | Dataset title | Dataset URL | Database and Identifier |
|---|---|---|---|---|
| Wang X | 2020 | | https://www.ncbi.nlm.nih.gov/nuccore/JACDXZ000000000 | NCBI GenBank, JACDXZ000000000 |
| Wang X | 2020 | | https://www.ncbi.nlm.nih.gov/sra/?term=prjna646786 | NCBI Sequence Read Archive, PRJNA646786 |
| Li CY, Zhang YZ | 2020 | | https://www.rcsb.org/structure/7CM9 | RCSB Protein Data Bank, 7CM9 |

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
