## [Decision Letter]

**Acceptance summary:**

Following the very interesting discussion with the referees and the resulting thorough revision I am happy to see this work now ready for publication in *eLife*. It will add to the growing knowledge of the marine sulfur cycle and maybe even stimulate follow-up work on the soil systems you mention. All in all I thank you for your additional effort you put into the work and I am looking forward to seeing it published.

**Decision letter after peer review:**

Thank you for submitting your article "Novel enzyme for dimethyl sulfide-releasing in bacteria reveals a missing route in the marine sulfur cycle" for consideration by *eLife*. Your article has been reviewed by 3 peer reviewers, including Geog Pohnert as the Reviewing Editor and Reviewer #1, and the evaluation has been overseen by Meredith Schuman as the Senior Editor.

The reviewers have discussed the reviews with one another and the Reviewing Editor has drafted this decision to help you prepare a revised submission.

Summary:

The study isolates bacteria from diverse Antarctic samples which utilise DMSP as the sole carbon source. It initially focuses on a Gammaproteobacterium, Psychrobacter sp.D2, which the authors establish lacks a known DMSP lyase enzyme despite having DMSP lyase activity (this needs to be quantified). Through RNA-seq and bioinformatics, they identify the gene cluster responsible for this activity and identify a novel DMSP lyase somewhat related to DddD in that it involves CoA, but critically also ATP, which distinguishes it from the pack of other known Ddd enzymes. This enzyme is a ATP-dependent DMSP CoA synthase required for growth on DMSP and its transcription is upregulated by DMSP availability. The novel mechanism of this enzyme is proposed from a strong structural component to the study. The authors propose the downstream pathway for DMSP catabolism, which we find to be oversold and requiring gene mutagenesis to confirm, and to be preliminary in comparison with the authors' other findings. Finally, the study attempts to show how widespread the enzyme is in sequenced bacteria, confidently showing it to be functional in other related Gammaproteobacteria and some Firmicutes.

Essential revisions:

1. Title: “a missing route" was it really missing? We would suggest a more precise title. Would be better to say "that releases DMS" or an alternative.

2. This is a Ddd enzyme by definition and should be named as such.

Line 27- We disagree with the use of a new gene prefix when there is a strong precedent for the use of Ddd for "DMSP-dependent DMS". If this enzyme is a DMSP lyase and is in bacteria then its naming should follow protocol and be called Ddd"X"-X-being a letter not currently utilised in known systems. Deviating from this convention causes confusion and is not appropriate. Furthermore, AcoD is already assigned in some bacteria to acetaldehyde dehydrogenase II.

3. As presented, the bioinformatics-based evidence regarding the broad distribution of this enzyme (as claimed e.g. in the Abstract, line 33) does not stand up.

Currently as presented in the manuscript, especially Figure 6, we are led to believe the enzyme is more widespread than can be demonstrated based on the authors' evidence (i.e., the authors allow a very low threshold of sequence identity and claim function outside of the groups they have tested). Either more work is needed to show that claims of such a wide distribution are merited, or the authors should limit their claims to what can be substantiated by their work. Specifically, the authors cannot comment on the "functional" enzyme being widespread outside of the Gamma's and Firmicutes that were tested, let alone the importance of the role in DMSP cycling. Only three "AcoD" enzymes were ratified in this study, which are relatively closely related to each (Psychrobacter sp. D2 Sporosarcina sp. P33 and Psychrobacter sp. P11G5 that are > 77% identical to each other). As can be seen in Figure 6, these three proteins cluster together and are far removed from all the other sequences on the figure, for which we have no evidence of their function (i.e., nothing can realistically be said on Deltas, Actinos or Alphas or the MAGS). Just to be clear, these other proteins shown in clades above and below the functional "AcoDs" in Figure 6 are only ~30% identical to ratified "AcoD".

Furthermore, only strain D2 was shown to make DMS; none of the other strains were tested. Far more testing of the diverse enzymes and strains are needed to make these statements as this study only tests one strain and three of the closely related enzymes (defined on Figure 6).

Additional specific comments on this issue:

Line 280. The sentence on MAGS and the environments containing them does not stand up for reasons summarised above. All MAGS shown on Figure 6 are not similar enough to "AcoD" to be termed as functional Ddd enzymes. More work has to be done on the strains and enzymes that are more divergent to true "AcoDs" before such a statement is supported. Please delete.

Line 509-We agree with what the authors write about stringency. However, these parameters do not seem to have been utilised as stated here. Their stringency statement holds up for comparison between the D2 "AcoD" and two other tested "AcoD" enzymes and all those in the middle clade on Figure 6. But this is not the case for the proteins shown above and below this "AcoD" clade in Figure 6 which have at best around 30% identity to characterised enzymes. See below for examples. As the authors state in their methods, high-stringency methods are needed to exclude other acetyl-CoA synthetase family proteins. Thus, most of the genes shown on fig6 cannot be taken as having this Ddd activity.

"To further validate that these AcoD homologs" the authors examined the activity of two closely related enzymes from a group of nine homologs with > 65 % sequence identity (starting line 283, Figure 6). It is not surprising that these enzymes have the same activity. Homologs outside this group of nine (Figure 6) are far less related to the characterized AcoD (< 32 % seq. identity). Conservation of the phosphate-transferring His (His292) and an active site Trp (Trp391) does not seem to be strong evidence for functional conservation. The manuscript does not provide any additional evidence that these less related enzymes also degrade DMSP. Either more experimentation is necessary, or the paragraph on the "Distribution of the ATP DMSP lysis pathway in bacteria" must be revised. For example:

Psychrobacter AcoD (WP_068035783.1) is 31% identical to Bilophila sp. 4_1_30 (WP_009381183.1) in the below group of bacteria on Figure 6.

Psychrobacter AcoD (WP_068035783.1) is 29% identical to Thermomicrobium roseum (WP_041435830.1) in the above group of bacteria on Figure 6.

Line 283. This is not the case! The two sequences that were chosen to "validate" are far to close to the D2 "AcoD" than to MAGS and other potential "AcoDs" shown above and below the functional Ddd clade on Figure 6. This section design is weak and does not lend weight to the expansiveness of this family. More work on the more diverse enzymes and bacteria is needed to support the authors claims. Please delete or study the activity of the more diverse strains and their candidate "AcoDs".

Figure 6. This is a nicely presented figure that unfortunately slightly deceives the reader. The authors need to clearly show which strains they have shown to have Ddd activity (currently one as I understand it) and which enzymes they have shown to have the appropriate activity (currently three closely related enzymes as I understand it). If I am not wrong these are all confined to the middle clade of Gammas and Firmicutes. These stand clearly apart form the other strains (above and below) which have not been studied and which are only ~ 30% Identical to "AcoD" at the protein level. This is not clear on the figure and definitely misleads in the abstract and throughout the manuscript.

4. We expect to see kinetics done on the new enzyme in line with what the authors have done in other related studies on Ddd and Dmd enzymes.

This is important to place the work in context with previously identified Ddd and Dmd enzymes, many of which have been analysed by these authors in previous publications. The characterization of the AcoD activity remains entirely qualitative. The authors only provide relative activities measured at a single substrate concentration. This data does not support the following statement: "Mutations of these two residues significantly decreased the enzymatic activities of AcoD, suggesting that these residues play important roles in stabilizing the DMSP-CoA intermediate" (l.223-225).

5. The manuscript does provide unambiguous evidence for the activity of AcoD and its function during growth on DMSP. On the other hand, the description of the "ATP DMSP lysis pathway" is less clear.

Transcriptomics analysis (Figure 2C) suggest that growth on DMSP upregulate the genes 1696 (BCCT), 1697 (AcoD), 1698 and 1699. The function of the third and fourth protein remain unclear (line 253). Instead, a reductase (AcuI) encoded somewhere else on the same genome was shown to transform the acryloyl-CoA to propionate-CoA. What was the transcription profile of acuI acuH in the RNA-seq? were they induced by growth on DMSP? Is the 1696-1697-1698-1699 gene cluster conserved? What is the function of 1698 and 1699? These questions are only relevant if the authors plan to maintain the claim of having identified a new pathway. This pathway prediction component is very weak and could be supplemented by KO mutagenesis of the dddCB and acuI. Without such work this is speculation and needs to be written as such.

6. Appropriate controls, units and quantification should be used:

Line 102- Please give a normalised value for the level of DMS produced from DMSP per time and protein/cells.

Figure 2.

A. One would expect to see a growth curve of D2 on DMSP compared to acrylate, a conventional carbon source (e.g. pyruvate, glycerol or succinate) and a no carbon control. As "AcoD" is predicted to ligate CoA to DMSP it would be good to know if the strain grows on acrylate. It might be predicted to have different properties to e.g. Halomonas which does grow on acrylate. At least a no carbon and conventional carbon source should definitely be included.

B. The units for this figure are not appropriate. It would be more appropriate to show the actual amount of DMS that is produced by the strain, ideally normalised to protein, cells or absorbance and time. Detail in the figure what the control is.

C. Would like to see error bars on this figure. Also would have been sensible to colour code these to match panel D.

Figure 3.

B and C. as with Figure 2 we need to see levels of DMS normalised to cells/protein and time.

Line 374- No controls. Please include these as detailed above. No carbon, conventional carbon source, acrylate?

Quantitative data supporting Supplementary Figure 12 would be helpful. After all this route would have to explain that the bacteria can use acrylate CoA as sole carbon source (or at least alternatives would have to be discussed). Is the identified activity sufficient for this task?

Line 388- This method is/should be quantitative. It is standard practice to report DMS production normalised to time and cells/protein. Here we are only given peak area.

[Editors' note: further revisions were suggested prior to acceptance, as described below.]

Thank you for resubmitting your work entitled "A novel ATP dependent dimethylsulfoniopropionate lyase in bacteria that releases dimethyl sulfide and acryloyl-CoA" for further consideration by *eLife*. Your revised article has been reviewed by two peer reviewers and the evaluation has been overseen by Meredith Schuman as the Senior Editor and a Reviewing Editor.

The manuscript has been improved but there are some remaining issues that need to be addressed, as outlined below:

Both previous referees consider the revisions to be entirely satisfactory and have no further concerns. Given that one of the earlier referees is now a co-author, we decided to include a new third expert previously not involved.

This additional referee raises a significant concern regarding the appropriateness of re-naming DddX from an Acetyl-CoA lyase to DMSP lyase, and suggests comparably simple experiments for clarification: "In light of the above, before changing the annotation of the DddX enzyme to DMSP lyase, the authors should test if DddX lost its ability to catalyse its original and primary activity, which is to ligate acetate or propionate to CoA in the presence of ATP. A substrate competition assay that tests the ability of acetate and propionate to inhibit DMSP lyase activity would also be useful. Finally, the ability of the mutation ΔDddX to inhibit growth on acetate or propionate compared with the wild type would also be important to test."

Further, the reviewer suggests additional critical discussion of the distribution of the new enzyme that could be included in a revised version.

Please address all comments of the review.

*Reviewer #3:*

In my opinion, the authors addressed all important concerns.

*Reviewer #4:*

This study will be of interest to researchers within the fields of marine microbiology and enzymology. The study identified and characterized a new DMSP lyase enzyme in an Antarctic marine bacterium. Together with the eight previously identified DMSP lyase families, this enzyme may play an important role in the ocean sulfur cycle. While an experiment to verify the substrate specificity of the enzyme is still lacking, the majority of the conclusions drawn from the study are justified, and the data analysis is thorough.

This paper describes the identification and characterization of the dimethylsulfoniopropionate (DMSP) lyase from the Antarctic marine bacterium Psychrobacter sp. D2. The authors show that using these enzymes, Psychrobacter can catabolize DMSP and use it as a sole carbon source while releasing the ubiquitous organosulfur gas dimethyl sulfide (DMS). DMS is a bioactive signalling molecule that plays a key role in the oceanic food web, it is also responsible for the formation of cloud condensation nuclei in the atmosphere and is the primary source of biogenic sulfur emitted from the ocean. The authors use a diverse set of tools from classic microbiology, molecular biology, enzymology and crystallography to identify and characterize the enzyme. The newly identified DMSP lyase belongs to the acetyl-CoA synthase family and uses CoA and ATP to transform DMSP to acryloyl-CoA and DMS. The discovery of another DMSP lyase family is surprising given that eight DMSP lyase families, belonging to four distinct superfamilies, had already been identified and are widespread in the marine environment.

Strengths:

The researchers provides unambiguous evidence that the newly identified enzyme, DddX, allows Psychrobacter sp. D2 to grow on DMSP as a sole carbon source while releasing DMS. They also present rigorous characterization of the enzyme that includes structural analysis, proposed mode of action and kinetic parameters. Finally, the researchers provide evidence that other DddX homologous genes, obtained from a few species within the alphaproteobacteria, gammaproteobacteria and firmicutes, can also catalyse this reaction. In summary, the authors support most of their claims with data, and their findings are relevant for the understanding of the role of bacteria in the oceanic sulfur cycle.

Weaknesses:

Although the paper has several strengths, further work is necessary to test the specificity of DddX. Since DddX shows strong structure and sequence similarity to acetyl-CoA synthase enzymes (ACS), and due to its low catalytic activity toward DMSP, its low kcat / KM values, and its presence in a non-functional operon and in the genomes of soil bacteria (where DMSP is not commonly present) the substrate specificity of DddX toward DMSP is still puzzling. Given that acetyl-CoA synthase enzymes have been shown to have promiscuous activity toward different short chain fatty acids including unnatural substrates (Patel et al. JBC, 1987), it is feasible that in this study the authors actually describe an acetyl-CoA synthase enzyme with a mild promiscuous activity toward DMSP. In light of the above, before changing the annotation of the DddX enzyme to DMSP lyase, the authors should test if DddX lost its ability to catalyse its original and primary activity, which is to ligate acetate or propionate to CoA in the presence of ATP. A substrate competition assay that tests the ability of acetate and propionate to inhibit DMSP lyase activity would also be useful. Finally, the ability of the mutation ΔDddX to inhibit growth on acetate or propionate compared with the wild type would also be important to test. The results from the proposed experiments would not change the main conclusion of the paper, that DddX can catalyse the formation of DMS. Yet it may have a significant impact on the interpretation of the finding.

To assess the generality of their finding, the authors should also provide further details about the diversity and abundance of the DddX enzyme in the oceans and in other environments. For example, it would be appropriate to discuss why DddX enzymes are found also in soil and other non-marine environments, where, according to current knowledge, DMSP is not commonly produced (note that DMSP is produced by many organisms including marine algae, marine bacteria and corals and can be found in abundance in the oceans, in salt marshes, sea sediments and in the roots of specialized DMSP-producing plants). Is DddX a "niche" enzyme that can operate in multiple specialized species? Or is it an enzyme that is present in many bacteria in the ocean? As it stands, additional clarification and discussion of those points is required in the text.

Comments for authors:

1. The authors show that, under the conditions tested in the lab, DddX clearly supports the growth of Psychrobacter sp. D2 on DMSP as a sole carbon source and the DddX enzyme is able to catalyze the conversion of DMSP to DMS. However, the relatively low catalytic activity of DddX, low kcat/KM values, and the appearance of DddX in non-functioning operon and in the genomes of soil bacteria, raise questions about DddX substrate specificity. Given that Acetyl-CoA synthase enzymes have been shown to have promiscuous activity toward different short chain fatty acids and even towards unnatural substrates (Smita S. Patel and David R. Walt. JBC, 1987), It is possible, and not unlikely, that the authors actually describe in their study an acetyl-CoA synthase enzyme with a mild promiscuous activity toward DMSP. In Fact from the partial alignment the authors present in Figure 4—figure supplement 3, DddX is extremely similar to other Acetyl-CoA synthase enzymes (ACS). And as mentioned by the authors, it is also very similar in structure, having all the characteristics and active sites residues of other ACS enzymes. In light of the above, I think it's important that the authors test if the enzyme lost its original activity, to ligate acetate or propionate to CoA in the presence of ATP, before changing DddX annotation from "ACS" to "DMSP lyase". A substrate competition assay that tests the ability of acetate and propionate to inhibit DMSP lyase activity would also be useful. And, testing the ability of the ΔDddX to inhibit growth on acetate or propionate and compare it with the growth of the wild-type would also be important.

If DddX lost its original function, then the authors should definitely use the name DddX and annotate their newly discovered enzyme as DMSP lyase. However, if not the author would probably want to keep the original name and maybe add a second name that includes the newly discovered function. Or at least discuss the possibility that this enzyme is bifunctional in their paper. The proposed set of experiments that we offer does not change the conclusion that DddX can catalyse the formation of DMS, but it may significantly change how we interpret the results.

Reference: Smita S Patel and David R. Walt. Substrate specificity of acetyl coenzyme A synthetase. JBC (1987) (https://doi.org/10.1016/S0021-9258(18)48214-2)

2. The authors should explain as well as discuss in the paper how DddX enzymes are found in environments such as soil and other non-marine environments, where, according to our knowledge, DMSP is not commonly produced. Note that DMSP is produced by many organisms including marine algae, marine bacteria and corals and can be found in abundance in the oceans, in salt marshes, sediments or in roots of specialized DMSP producing plants but is it also produce in other soil environments? Please provide a brief description in the text of where the bacteria with DddX in their genome can be found. Also in Figure 6, please indicate the general location of isolation near each branch as in the previous version of the figure. Please search for it for each genome in the NCBI bioSample description. For example: Sporosarcina sp. P33 (with functional DMSP lyase) is a soil bacteria – (The location of isolation is interesting) – https://www.ncbi.nlm.nih.gov/biosample/SAMN04589807

3. Please explain in more detail the general abundance of the DddX enzyme in the oceans (or other environments). Please use the "Tara oceans gene atlas" or other tools to support the high/low abundance of the DddX enzyme in the environment. Is DddX a "niche" enzyme that can operate in multiple specialized species and locations? Or is it one of the enzymes that many bacteria in the ocean possess? As it stands now it is not clear from the text/data. For example: In Line 36 – 38 the authors claims that "DddX is found in diverse marine alphaproteobacteria, gammaproteobacteria and firmicutes, suggesting that this new DMSP lyase may play an important role in DMSP/DMS cycles" – However, in Figure 6 It appears that there are not many organisms belonging to those diverse groups of bacteria. i.e. There are a very large number of bacterial species between and within alphaproteobacteria, gammaproteobacteria and firmicutes that do not have this gene. In fact from the data presented, DddX enzymes can be found sporadically in only very few groups of species which are very distinct from one another. In summary, try to be more specific in the description of the enzyme diversity and abundance throughout the paper. If DddX is only a "niche" DMSP lyase, the study is still very interesting and valid. However, one would not want to give the wrong impression about the abundance of a newly discovered enzyme. Make sure to discuss the abundance of DddX and the apparently sporadic diversity of the DddX enzyme in the discussion of the paper.

---

## [Author Response]

Essential revisions:1. Title: “a missing route" was it really missing? We would suggest a more precise title. Would be better to say "that releases DMS" or an alternative.

We have changed the title to "A novel ATP dependent dimethylsulfoniopropionate lyase in bacteria that releases dimethyl sulfide and acryloyl-CoA" following the reviewers’ comment.

2. This is a Ddd enzyme by definition and should be named as such.Line 27- We disagree with the use of a new gene prefix when there is a strong precedent for the use of Ddd for "DMSP-dependent DMS". If this enzyme is a DMSP lyase and is in bacteria then its naming should follow protocol and be called Ddd"X"-X-being a letter not currently utilised in known systems. Deviating from this convention causes confusion and is not appropriate. Furthermore, AcoD is already assigned in some bacteria to acetaldehyde dehydrogenase II.

We agree with the reviewers that this novel enzyme is a Ddd enzyme. The letter “X” has not been utilized in the reported systems and thus we have changed the name of this enzyme to DddX.

3. As presented, the bioinformatics-based evidence regarding the broad distribution of this enzyme (as claimed e.g. in the Abstract, line 33) does not stand up.Currently as presented in the manuscript, especially Figure 6, we are led to believe the enzyme is more widespread than can be demonstrated based on the authors' evidence (i.e., the authors allow a very low threshold of sequence identity and claim function outside of the groups they have tested). Either more work is needed to show that claims of such a wide distribution are merited, or the authors should limit their claims to what can be substantiated by their work. Specifically, the authors cannot comment on the "functional" enzyme being widespread outside of the Gamma's and Firmicutes that were tested, let alone the importance of the role in DMSP cycling. Only three "AcoD" enzymes were ratified in this study, which are relatively closely related to each (Psychrobacter sp. D2 Sporosarcina sp. P33 and Psychrobacter sp. P11G5 that are > 77% identical to each other). As can be seen in Figure 6, these three proteins cluster together and are far removed from all the other sequences on the figure, for which we have no evidence of their function (i.e., nothing can realistically be said on Deltas, Actinos or Alphas or the MAGS). Just to be clear, these other proteins shown in clades above and below the functional "AcoDs" in Figure 6 are only ~30% identical to ratified "AcoD".Furthermore, only strain D2 was shown to make DMS; none of the other strains were tested. Far more testing of the diverse enzymes and strains are needed to make these statements as this study only tests one strain and three of the closely related enzymes (defined on Figure 6).Additional specific comments on this issue:Line 280. The sentence on MAGS and the environments containing them does not stand up for reasons summarised above. All MAGS shown on Figure 6 are not similar enough to "AcoD" to be termed as functional Ddd enzymes. More work has to be done on the strains and enzymes that are more divergent to true "AcoDs" before such a statement is supported. Please delete.Line 509-We agree with what the authors write about stringency. However, these parameters do not seem to have been utilised as stated here. Their stringency statement holds up for comparison between the D2 "AcoD" and two other tested "AcoD" enzymes and all those in the middle clade on Figure 6. But this is not the case for the proteins shown above and below this "AcoD" clade in Figure 6 which have at best around 30% identity to characterised enzymes. See below for examples. As the authors state in their methods, high-stringency methods are needed to exclude other acetyl-CoA synthetase family proteins. Thus, most of the genes shown on fig6 cannot be taken as having this Ddd activity."To further validate that these AcoD homologs" the authors examined the activity of two closely related enzymes from a group of nine homologs with > 65 % sequence identity (starting line 283, Figure 6). It is not surprising that these enzymes have the same activity. Homologs outside this group of nine (Figure 6) are far less related to the characterized AcoD (< 32 % seq. identity). Conservation of the phosphate-transferring His (His292) and an active site Trp (Trp391) does not seem to be strong evidence for functional conservation. The manuscript does not provide any additional evidence that these less related enzymes also degrade DMSP. Either more experimentation is necessary, or the paragraph on the "Distribution of the ATP DMSP lysis pathway in bacteria" must be revised. For example:Psychrobacter AcoD (WP_068035783.1) is 31% identical to Bilophila sp. 4_1_30 (WP_009381183.1) in the below group of bacteria on Figure 6.Psychrobacter AcoD (WP_068035783.1) is 29% identical to Thermomicrobium roseum (WP_041435830.1) in the above group of bacteria on Figure 6.Line 283. This is not the case! The two sequences that were chosen to "validate" are far to close to the D2 "AcoD" than to MAGS and other potential "AcoDs" shown above and below the functional Ddd clade on Figure 6. This section design is weak and does not lend weight to the expansiveness of this family. More work on the more diverse enzymes and bacteria is needed to support the authors claims. Please delete or study the activity of the more diverse strains and their candidate "AcoDs".Figure 6. This is a nicely presented figure that unfortunately slightly deceives the reader. The authors need to clearly show which strains they have shown to have Ddd activity (currently one as I understand it) and which enzymes they have shown to have the appropriate activity (currently three closely related enzymes as I understand it). If I am not wrong these are all confined to the middle clade of Gammas and Firmicutes. These stand clearly apart form the other strains (above and below) which have not been studied and which are only ~ 30% Identical to "AcoD" at the protein level. This is not clear on the figure and definitely misleads in the abstract and throughout the manuscript.

Re DddX identity amino acid homology issues, we thank the reviewers for their comments and apologize for the confusion. The sequence identity cut-off was set at >30% (The 50% cut-off value was previously used to investigate the distribution of conserved ATP-grasp domain in the database in an early version of the manuscript; this error has now been corrected in this revision). After retrieving putative DddX homologues from the NCBI ref-seq database, multiple sequence alignment was manually inspected to search for the presence of conserved residues that we predict to be involved in DddX catalysis (e.g. H292, W391 and D432).

Re MAGS low homology issues, we fully agree with the reviewers on the inclusion of MAG sequences with lower identity to ratified DddX. For that reason, we have removed all mention of MAG analysis from Figure 6 and the manuscript. We thank the reviewers for pointing out these problems and feel the manuscript is tighter without their inclusion.

Re validation issues, we thank the reviewers for revealing the limitations of our initial analysis. Following the reviewers’ advice, we now include new data that validate representative DddX from the more divergent alphaproteobacterial and gammaproteobacterial DddX homologs (with only ~30% identity to *Psychrobacter* DddX). We chemically synthesized the *dddX* sequences from alphaproteobacterial *Pelagicola* sp. LXJ1103 and gammaproteobacterial *Marinobacterium jannaschii*, overexpressed them in *E. coli*, purified the proteins and carried out enzyme assays. Both these candidate DddX enzymes had the predicted DMSP lysis activity producing DMS and acryloyl-CoA (*Figure 6—figure supplement 1).* As covered above, the homology of DddX proteins to *Psychrobacter* DddX is now clearly shown on the new Figure 6 as is the representatives that were shown to be functional. We have also thoroughly edited the manuscript throughout to convey the real known scope of DddX. Although we could have synthesised and assayed more DddX representatives, we now feel that the spread of validation is suitable for what we want to convey.

Re strain assays, we realise that it would have been more complete assay the strains that contain the more diverse functional DddX enzymes to confirm their activity. However, in these times it was difficult to swiftly procure suitable strains. We hope that the reviewers agree with us that the functional ratification of the enzymes is sufficient for publication here.

4. We expect to see kinetics done on the new enzyme in line with what the authors have done in other related studies on Ddd and Dmd enzymes.This is important to place the work in context with previously identified Ddd and Dmd enzymes, many of which have been analysed by these authors in previous publications. The characterization of the AcoD activity remains entirely qualitative. The authors only provide relative activities measured at a single substrate concentration. This data does not support the following statement: "Mutations of these two residues significantly decreased the enzymatic activities of AcoD, suggesting that these residues play important roles in stabilizing the DMSP-CoA intermediate" (l.223-225).

We thank the reviewers for pointing out this omission. We have now measured the enzymatic properties including the kinetic parameters of DddX from *Psychrobacter* sp. D2. DddX exhibited a *K_m_* value of 0.4 mM for DMSP, which is lower than DmdA and most other known DMSP lyases. The kinetic data of DddX (Figure 3—figure supplement 3), comparisons to other related enzymes (Supplementary file 1c), and the methods used are now added to the revised manuscript.

We agree with the reviewers that our current data does not support the roles of the residues Trp391 and Phe435 in stabilizing the DMSP-CoA intermediate. Because DMSP-CoA is an unstable intermediate, and it is difficult to measure the kinetic parameters of DddX mutants towards DMSP-CoA, we have now modified this sentence to "Mutations of these two residues significantly decreased the enzymatic activities of DddX, suggesting that these residues play important roles in DddX catalysis" in line 237-239 in the revised manuscript.

5. The manuscript does provide unambiguous evidence for the activity of AcoD and its function during growth on DMSP. On the other hand, the description of the "ATP DMSP lysis pathway" is less clear.Transcriptomics analysis (Figure 2C) suggest that growth on DMSP upregulate the genes 1696 (BCCT), 1697 (AcoD), 1698 and 1699. The function of the third and fourth protein remain unclear (line 253). Instead, a reductase (AcuI) encoded somewhere else on the same genome was shown to transform the acryloyl-CoA to propionate-CoA. What was the transcription profile of acuI acuH in the RNA-seq? were they induced by growth on DMSP? Is the 1696-1697-1698-1699 gene cluster conserved? What is the function of 1698 and 1699? These questions are only relevant if the authors plan to maintain the claim of having identified a new pathway. This pathway prediction component is very weak and could be supplemented by KO mutagenesis of the dddCB and acuI. Without such work this is speculation and needs to be written as such.

We thank the reviewers for the insightful comments and agree that future experiments are required in order to firmly establish the pathway of acryloyl-CoA degradation. We have now toned down the description of the pathway and moved the related text into the Discussion section in line 315-324 in the revised manuscript, which reads:

“If *Psychrobacter* sp. D2 employs its AcuH homolog to convert acryloyl-CoA to 3-HP-CoA (Cao et al., 2017),then, given the high sequence identity of 1698 to DddC and 1699 to DddB, it is possible that these enzymes further catabolize 3-HP-CoA to acetyl-CoA (Alcolombri et al., 2014; Curson et al., 2011b). […] Thus, *Psychrobacter* sp. D2 may also employ an AcuI homolog (i.e. 0105) to convert acryloyl-CoA to propionate-CoA, which would be metabolized through the methylmalonyl-CoA pathway (Reisch et al., 2013).”

6. Appropriate controls, units and quantification should be used:Line 102- Please give a normalised value for the level of DMS produced from DMSP per time and protein/cells.Figure 2.A. One would expect to see a growth curve of D2 on DMSP compared to acrylate, a conventional carbon source (e.g. pyruvate, glycerol or succinate) and a no carbon control. As "AcoD" is predicted to ligate CoA to DMSP it would be good to know if the strain grows on acrylate. It might be predicted to have different properties to e.g. Halomonas which does grow on acrylate. At least a no carbon and conventional carbon source should definitely be included.

Thank you for these important suggestions which we have now fully taken on. We have now added the quantitative data and appropriate controls in Figure 2 in the manuscript according to the reviewers’ suggestions. The growth curves of *Psychrobacter* sp. D2 on sodium pyruvate and acrylate, and a no carbon control are now added in Figure 2A. *Psychrobacter* sp. D2 could not use acrylate as a sole carbon source for growth in contrast to *Halomonas*, which is discussed in the discussion now in line 325-334.

B. The units for this figure are not appropriate. It would be more appropriate to show the actual amount of DMS that is produced by the strain, ideally normalised to protein, cells or absorbance and time. Detail in the figure what the control is.We are sorry but feel it is important to show the chromatogram for DMS against a standard and a now clearly labelled media control. However, we do now report the amount of DMS that is generated normalised to protein levels both in the main text (line 106) and in the figure legend for Figure 2B. Sorry for our previous oversight.C. Would like to see error bars on this figure. Also would have been sensible to colour code these to match panel D.Figure 3.B and C. as with Figure 2 we need to see levels of DMS normalised to cells/protein and time.Line 374- No controls. Please include these as detailed above. No carbon, conventional carbon source, acrylate?Quantitative data supporting Supplementary Figure 12 would be helpful. After all this route would have to explain that the bacteria can use acrylate CoA as sole carbon source (or at least alternatives would have to be discussed). Is the identified activity sufficient for this task?Line 388- This method is/should be quantitative. It is standard practice to report DMS production normalised to time and cells/protein. Here we are only given peak area.

Thanks for these sensible suggestions. We have now coordinated the colour of Figure 2C and D. This is much improved for this. Re the error bars, *RNA sequencing was carried out on a pool of* three biological samples to produce a single library. We thus could not provide error bars here. However, the upregulation of these genes was confirmed by RT-qPCR that was done in triplicate. Thus, instead of the RNA seq data we include the RT-qPCR data in Figure 2C with error bars. We feel that this is much more appropriate. As covered above, DMS levels are reported appropriately normalised in Figure 3.

Furthermore, we acknowledge that the genes and metabolic pathway involved in acryloyl-CoA metabolism in this bacterium is unclear and requires further work. The amount of propionate-CoA normalised to protein levels is reported in the figure legend for *Figure 1—figure supplement 2* (supplementary Figure 12 in the original version of manuscript). We now discuss several possibilities in line 312-324 in the revised version of manuscript.

[Editors' note: further revisions were suggested prior to acceptance, as described below.]

Reviewer #4:[…] Comments for authors:1. The authors show that, under the conditions tested in the lab, DddX clearly supports the growth of Psychrobacter sp. D2 on DMSP as a sole carbon source and the DddX enzyme is able to catalyze the conversion of DMSP to DMS. However, the relatively low catalytic activity of DddX, low kcat/KM values, and the appearance of DddX in non-functioning operon and in the genomes of soil bacteria, raise questions about DddX substrate specificity. Given that Acetyl-CoA synthase enzymes have been shown to have promiscuous activity toward different short chain fatty acids and even towards unnatural substrates (Smita S. Patel and David R. Walt. JBC, 1987), It is possible, and not unlikely, that the authors actually describe in their study an acetyl-CoA synthase enzyme with a mild promiscuous activity toward DMSP. In Fact from the partial alignment the authors present in Figure 4—figure supplement 3, DddX is extremely similar to other Acetyl-CoA synthase enzymes (ACS). And as mentioned by the authors, it is also very similar in structure, having all the characteristics and active sites residues of other ACS enzymes. In light of the above, I think it's important that the authors test if the enzyme lost its original activity, to ligate acetate or propionate to CoA in the presence of ATP, before changing DddX annotation from "ACS" to "DMSP lyase". A substrate competition assay that tests the ability of acetate and propionate to inhibit DMSP lyase activity would also be useful. And, testing the ability of the ΔDddX to inhibit growth on acetate or propionate and compare it with the growth of the wild-type would also be important.If DddX lost its original function, then the authors should definitely use the name DddX and annotate their newly discovered enzyme as DMSP lyase. However, if not the author would probably want to keep the original name and maybe add a second name that includes the newly discovered function. Or at least discuss the possibility that this enzyme is bifunctional in their paper. The proposed set of experiments that we offer does not change the conclusion that DddX can catalyse the formation of DMS, but it may significantly change how we interpret the results.Reference: Smita S Patel and David R. Walt. Substrate specificity of acetyl coenzyme A synthetase. JBC (1987) (https://doi.org/10.1016/S0021-9258(18)48214-2)

Re the substrate specificity of DddX, we thank the reviewer for the insightful comments. We tested the enzymatic activities of recombinant DddX towards acetate and propionate. While DddX converted DMSP to DMS and acryloyl-CoA, it exhibited no activity towards acetate or propionate (Figure 3—figure supplement 4).

Re the substrate competition assay, we tested the enzymatic activity of DddX towards DMSP in the presence of acetate or propionate. Acetate or propionate (with a final concentration of up to 5 mM) had little effect on the enzymatic activity of DddX towards DMSP (Figure 3—figure supplement 5).

Re strain growth experiment, we tested the ability of the wild-type strain D2 and its mutants to grow with acetate or propionate as the sole carbon source. The wild-type strain D2 could use acetate or propionate as the sole carbon source, and the deletion of *dddX* has little effect on its growth on acetate or propionate (*Figure 3—figure supplement 6*), suggesting that *dddX* is unlikely to be involved in acetate and propionate catabolism in strain D2.

The results above also indicate that DddX does not use acetate nor propionate as a substrate, as such it does not function as an acetate-CoA ligase. We therefore propose to keep the name “DddX” in the manuscript.

2. The authors should explain as well as discuss in the paper how DddX enzymes are found in environments such as soil and other non-marine environments, where, according to our knowledge, DMSP is not commonly produced. Note that DMSP is produced by many organisms including marine algae, marine bacteria and corals and can be found in abundance in the oceans, in salt marshes, sediments or in roots of specialized DMSP producing plants but is it also produce in other soil environments? Please provide a brief description in the text of where the bacteria with DddX in their genome can be found. Also in Figure 6, please indicate the general location of isolation near each branch as in the previous version of the figure. Please search for it for each genome in the NCBI bioSample description. For example: Sporosarcina sp. P33 (with functional DMSP lyase) is a soil bacteria – (The location of isolation is interesting) – https://www.ncbi.nlm.nih.gov/biosample/SAMN04589807

We thank this reviewer for the comments and we have now added back the location/environment of isolation for the strains shown in the phylogenetic tree (see revised Figure 6). It is difficulty to speculate the source of DMSP in soils since this has not been studied extensively. However, it is likely that soil-dowelling microbes may benefit from DMSP released from plants. We added the following sentence in the discussion, which reads:

“Interestingly, DddX is found in several bacterial isolates which were isolated from soil or plant roots, suggesting that DMSP may also be produced in these ecosystems”.

3. Please explain in more detail the general abundance of the DddX enzyme in the oceans (or other environments). Please use the "Tara oceans gene atlas" or other tools to support the high/low abundance of the DddX enzyme in the environment. Is DddX a "niche" enzyme that can operate in multiple specialized species and locations? Or is it one of the enzymes that many bacteria in the ocean possess? As it stands now it is not clear from the text/data. For example: In Line 36 – 38 the authors claims that "DddX is found in diverse marine alphaproteobacteria, gammaproteobacteria and firmicutes, suggesting that this new DMSP lyase may play an important role in DMSP/DMS cycles" – However, in Figure 6 It appears that there are not many organisms belonging to those diverse groups of bacteria. i.e. There are a very large number of bacterial species between and within alphaproteobacteria, gammaproteobacteria and firmicutes that do not have this gene. In fact from the data presented, DddX enzymes can be found sporadically in only very few groups of species which are very distinct from one another. In summary, try to be more specific in the description of the enzyme diversity and abundance throughout the paper. If DddX is only a "niche" DMSP lyase, the study is still very interesting and valid. However, one would not want to give the wrong impression about the abundance of a newly discovered enzyme. Make sure to discuss the abundance of DddX and the apparently sporadic diversity of the DddX enzyme in the discussion of the paper.

We thank the reviewer for this comment. We have focused our analysis of the distribution of DddX in genome sequenced bacteria but not environmental omics datasets. As pointed out previously by other reviewers, it is necessary to test the function of those homologous sequences retrieved from environmental metagenomics datasets in order to ascertain their role in DMSP degradation. This proves challenging since those homologs returned from Tara Oceans datasets are usually short and do not always contain the full open reading frame, making it difficult for gene synthesis and overexpression in *E. coli*. In agreeance with the comments from other reviewers, we have focused on our analysis of DddX in genome-sequence bacteria isolates instead, and have chemically-synthesize several of these homologs to validate their role in DMSP catabolism (Figure 6). We have now modified the following sentences in the abstract, which reads:

“DddX is found in several Alphaproteobacteria, Gammaproteobacteria and Firmicutes, suggesting that this new DMSP lyase may play an overlooked role in DMSP/DMS cycles.”

We also revised the last paragraph of discussion to reflect the fact that DddX is found in several bacterial isolates.